# Shannon entropy of transport self-organization due to dissolution–precipitation reaction at varying Peclet numbers in initially homogeneous porous media

**Evgeny Shavelzon and Yaniv Edery**

Technion – Israel Institute of Technology, Haifa, Israel

**Correspondence:** Evgeny Shavelzon (eshavelzon@campus.technion.ac.il)

**Abstract.** Dissolution and precipitation processes in reactive transport in porous media are ubiquitous in a multitude of contexts within the field of Earth sciences. In particular, the dynamic interaction between the reactive dissolution and precipitation processes and the solute transport is of interest as it is capable of giving rise to the emergence of preferential flow paths in the porous host matrix. It has been shown that the emergence of preferential flow paths can be considered to be a manifestation of transport self-organization in porous media as these create spatial gradients that distance the system from the state of perfect mixing and allow for a faster and more efficient fluid transport through the host matrix. To investigate the dynamic feedback between the transport and the reactive processes in the field and its influence on the emergence of transport self-organization, we consider a two-dimensional Darcy-scale formulation of a reactive-transport setup, where the precipitation and dissolution of the host matrix are driven by the injection of an acid compound, establishing local equilibrium with the resident fluid and an initially homogeneous porous matrix, composed of a calcite mineral. The coupled reactive process is simulated in a series of computational analyses employing the Lagrangian particle-tracking (LPT) approach, capable of capturing the subtleties of the multiple-scale heterogeneity phenomena. We employ the Shannon entropy to quantify the emergence of self-organization in the field, which we define as a relative reduction in entropy compared to its maximum value. Scalability of the parameters, which characterize the evolution of the reactive process, with the Peclet number in an initially homogeneous field is derived using a simple one-dimensional ADRE model with a linear adsorption reaction term and is then confirmed through numerical simulations, with the global reaction rate, the mean value, and the variance of the hydraulic-conductivity distribution in the field all exhibiting dependency on the reciprocal of the Peclet number. Our findings show that transport self-organization in an initially homogeneous field increases with time, along with the emergence of the field heterogeneity due to the interaction between the transport and reactive processes. By studying the influence of the Peclet number on the reactive process, we arrive at a conclusion that self-organization is more pronounced in diffusion-dominated flows, characterized by small Peclet values. The self-organization of the breakthrough curve exhibits the opposite tendencies, which are observed from the perspective of a thermodynamic analogy. The hydraulic power, required to maintain the driving head pressure difference between the inlet and outlet of the field, was shown to increase with the increasing variance, as well as with the increasing mean value of the hydraulic-conductivity distribution in the field, using a simple analytic model. This was confirmed by numerical experiments. This increase in power, supplied to the flow in the field, results in an increase in the level of transport self-organization. Employing a thermodynamic framework to investigate the dynamic reaction–transport interaction in porous media may prove to be beneficial whenever the need exists to establish relations between the intensification of the preferential flow path phenomenon, represented by a decline in the Shannon entropy of the transport, with the amount of reaction that occurred in the porous medium and the change in its heterogeneity.

# 1 Introduction

## 1.1 Dissolution–precipitation reaction and preferential flow paths in porous media

Dissolution and precipitation processes in reactive transport in porous media play an important role in multiple contexts in the field of Earth sciences, such as geological $CO_2$ storage (Ajayi et al., 2014), reactive contaminant transport (Brusseau, 1994), and acid injection in petroleum reservoirs (Shazly, 2021). They are responsible for the alteration of the transport characteristics of the porous media and for the emergence of preferential flow paths (Singurindy and Berkowitz, 2003; Al-Khulaifi et al., 2017) as dissolution–precipitation reactive processes lead to changes in the dimensions of the pores and their connectivity, thus introducing coupling between the chemical reaction and the transport in the media. This is observed, for example, in reactive infiltration in a porous medium, where the chemical reaction at the solid–fluid interface causes dissolution of the surrounding porous matrix, creating nonlinear feedback mechanisms that often lead to greatly enhanced permeability (Ladd and Szymczak, 2021; Szymczak and Ladd, 2006). The solid–fluid interface instability, observed in this case, is similar to the viscous-fingering phenomenon in the field of multi-phase flow, with undulation areas being formed first at the solid–fluid interface of the porous media due to dissolution and then later being transformed into well-defined, finger-like channels or wormholes that rapidly advance into the medium. As dissolution proceeds, these fingers interact, competing for the available flow.

These wormholes that funnel the flow can be regarded as preferential flow paths, ubiquitous in heterogeneous porous media, where most of the transport is concentrated. The importance of preferential flow paths in determining the transport properties of the porous media is widely recognized as they allow rapid solute transport and alter residence times (Beven and Germann, 1982; Stamm et al., 1998; Radolinski et al., 2022; Edery et al., 2014, 2016a). As these preferential flow paths align with the average flow direction, they introduce solute concentration gradients in the direction transverse to flow (Zehe et al., 2021) and an increasingly non-Fickian transport behavior, thus requiring a global homogenization approach for transport characterization that considers these subtleties in the upscaling and volume-averaging methods. The formation of preferential flow paths in the porous matrix due to the interaction between dissolution reaction and transport processes has been reported in a multitude of experimental studies (Kamolpornwijita et al., 2003; Zhang et al., 2021; Snippe et al., 2020; Li et al., 2019), while the opposite scenario of a precipitation reaction, such as in the formation of carbonate deposits during $CO_2$ sequestration, was investigated by Yan et al. (2020); they reported clogging of pores, leading to the alteration or complete blockade of the preferential flow paths. As opposed to the case of highly conductive networks, such as fractured rock formations, the preferential flow paths in the saturated heterogeneous porous media are not always well defined. For this case, Zehe et al. (2021) have reported that a higher variance in the hydraulic-conductivity field leads to the intensification of the preferential flow path phenomenon via a stronger concentration of solute within a smaller number of paths. Changing the observing perspective, Berkowitz and Zehe (2020) have argued that the emergence of preferential flow paths can be viewed as a manifestation of self-organization as the spatial concentration gradients, created within the system, distance the system from the state of perfect mixing, thus allowing for a faster and more efficient fluid transport.

## 1.2 Self-organization in physical systems and thermodynamic framework

The findings discussed above have brought us closer to classification of the emerging preferential flow paths in reactive transport in porous media as an embodiment of self-organization. Before addressing the specific problem of the emergence of preferential flow paths, it is in order to first discuss some basic concepts related to the phenomenon of self-organization in physical systems. Self-organization refers to a broad range of pattern formation processes, occurring through interactions internal to the system without intervention by external directing influences (Camazine et al., 2001). Examples of self-organization in physical systems include the formation of patterns in chemical reactions (Turing, 1952); animal behavior, such as bird flocking (Hemelrijk and Hildenbrandt, 2012); and river network formation (Stolum, 1996). In this context, it is interesting to consider reaction–diffusion systems, which have attracted much interest as a prototype model for pattern formation in nature. An attempt to provide an explanation of morphogenesis in the field of mathematical biology was first made by Alan Turing (Turing, 1952), who proposed a model based on chemical reaction–diffusion interaction, where two homogeneously distributed substances interact to produce stable, stationary patterns, represented by concentration deviations of the two substances inside the domain of the reaction. The reaction is autocatalytic and is represented by a system of nonlinear differential equations, thus possessing the capability for exhibiting self-organizing modes. The necessary condition for pattern emergence is that the diffusion coefficients of both substances differ, otherwise no self-organization will take place. This feature counteracts the homogenizing action of diffusion. Turning to the field of hydrology, Berkowitz and Zehe (2020) have stated that, while self-organization in well-defined networks, such as rivers, is clearly distinguishable, it is nevertheless also the case that less well-defined networks, such as transport in heterogeneous groundwater systems in the absence of fractures, are capable of displaying some characteristics of self-organization, which is manifested through spatially cor-

related, anisotropic patterns of structural and hydraulic properties.

Within the thermodynamic framework, Haken (1983) has defined self-organization as an emergence of ordered macroscale states in an open, complex system far from thermodynamic equilibrium due to the exchange of energy and matter with surroundings as a result of a synergetic interplay of microscale, irreversible processes. Recall the definition of entropy by Boltzmann (Sharp and Matschinsky, 2015), which is directly related to the number of possible microstates of a system that are consistent with a given macrostate, characterized by the macroscopic thermodynamic properties of the system (Haken, 1983). An ordered state is characterized by a reduction in the entropy of the system compared to its maximum value at equilibrium; thus, a macrostate with a lower amount of corresponding possible microstates is considered to be more ordered (Kondepudi and Prigogine, 1998). Since, according to the second law of thermodynamics, the overall entropy of the system and its surroundings cannot decrease, this reduction in entropy has to be exported from the system outside, leading to an increase in the entropy of its surroundings. The export of entropy from the system requires that physical work is performed on the system in order to maintain the current level of self-organization. One of the most exciting examples that comes to mind to illustrate this concept is biological life. Erwin Schrodinger, in his highly influential book *What is Life?* (Schrodinger, 1944), describes a living organism as a non-equilibrium thermodynamic system in interaction with its surroundings that maintains a level of inner self-organization. How does a living organism manage to avoid decay into the inert state of equilibrium? The answer is simple: by consuming food and transforming it into energy through metabolic processes. This energy allows the organism to maintain its level of self-organization by exporting the entropy, produced by irreversible physiological processes, to the surroundings. In other words, in order to maintain self-organization within itself, a living organism it works to decrease the level of self-organization of its surroundings by exporting entropy outside.

While the thermodynamic framework refers to physical entropy, as introduced by Clausius (1857), the information entropy, first defined by Shannon (1948), has proven to be extremely useful in quantifying self-organization in various fields of physical science. This parameter, also referred to as the Shannon entropy, was introduced originally in the framework of communication. Viewing communication as a statistical process, Shannon employed an entropy-like parameter, defined similarly to Boltzmann's definition of entropy, to provide a measure for the amount of transmitted information carried by a certain sequence of symbols. The applications of Shannon entropy in the field of geophysical science are plenty; Chiogna and Rolle (2017) have employed Shannon entropy to quantify dilution and reactive mixing in solute transport problems; Schweizer et al. (2017) used it for

uncertainty assessment in complex geological models, Mays et al. (2002) measured the temporal and spatial complexity of unsaturated flow in heterogeneous media, Woodbury and Ulrych (1996) applied the minimum relative entropy principle (MRE) from the field of information theory to the problem of recovering the release history of a groundwater contaminant, and Hansen et al. (2018, 2023) employed the Jaynes' maximum entropy principle to determine the flow configuration for the case of immiscible and incompressible two-phase flow in a porous medium. In addition, Zehe et al. (2021) studied self-organization in preferential flow paths in porous media with various degrees of spatial heterogeneity and quantified it in terms of Shannon entropy. They found that stronger transversal concentration gradients emerge with an increasing variance of the hydraulic-conductivity field, which is reflected in a smaller entropy of the transversal distribution of transport pathways. Their findings suggest that a higher variance of the hydraulic-conductivity field coincides with stronger self-organization of transport pathways (represented by steepening of the concentration gradient in the direction transverse to flow). The explanation to this non-intuitive finding is given in terms of energy: in the thermodynamic framework, the emergence of spatial self-organization requires energy input into the system, which grows along with the increase in the level of self-organization. Since the energy input in the form of hydraulic power, necessary to sustain steady-state fluid flow and tracer transport, grows with the variance of the hydraulic-conductivity field, this enables an increase in the self-organization of the transport in the field.

## 1.3 Lagrangian particle approach to transport in porous media

Among the family of numerical methods employed to simulate transport and reaction problems in porous media, the Lagrangian particle approach continues to gain more prominence in the recent decades, along with the continuing development of powerful computers (Jiao et al., 2021; Sole-Mari and Fernandez-Garcia, 2018; Sole-Mari et al., 2020). The Lagrangian family of computational methods considers the fluid phase to consist of discrete particles and tracks the path of each particle, as opposed to the more conventional Eulerian family of methods, where the fluid phase is treated as a continuum and where its governing equations are developed based on conservation principles (Zhang and Chen, 2007; Meakin and Tartakovsky, 2009). In the Lagrangian approach, motion and interaction of particles are defined by a specific set of laws that correspond to the physical situation of interest. Statistical analysis is then applied to the particle trajectories and interactions. The Lagrangian particle methods are not susceptible to the numerical instabilities that are often present in the numerical simulations where an Eulerian approach is employed (Meakin and Tartakovsky, 2009). Perez et al. (2019) employed random-walk particle tracking to simulate bimolecular, irreversible chemical re-

actions in porous media and have shown the equivalence of their formulation to the well-known advection–diffusion reaction equation (ADRE), while others used it to simulate reactive transport in heterogeneous fields that display non-Fickian behavior (Berkowitz et al., 2013; Edery et al., 2013, 2010, 2009, 2016b). Other applications of Lagrangian methods include particle dispersion in fluid (Shirolkar et al., 1996); multiphase flows (Meakin and Tartakovsky, 2009); reactive transport (Schmidt et al., 2019); and impact problems, such as water impingement on a surface (Petrosino et al., 2021).

### 1.4 Objectives

To investigate the dynamic interaction between the transport and the reactive process in an initially homogeneous porous medium and its influence on the emergence of transport self-organization in the medium, we consider a two-dimensional Darcy-scale formulation of a reactive-transport setup, where precipitation and dissolution in the medium are driven by the injection of an acid compound, establishing local equilibrium with the resident fluid and an initially homogeneous porous medium, composed of calcite mineral. The coupled reactive process is simulated in a series of computational analyses where the low-pH water is injected into an initially homogeneous domain, at first in equilibrium with the resident fluid (high-pH water). We employ a Lagrangian particle-tracking (LPT) approach, capable of capturing the subtleties of the multiple-scale heterogeneity phenomena, along the lines of Edery et al. (2021).

In particular, we are interested in the influence of the transport Peclet number on the reactive process and the emergence of self-organization of the transport in the porous medium. To investigate this relation, we simulate a number of reactive-transport scenarios for different values of advective to diffusive transport rates, characterized by the Peclet number. This is achieved by applying different values of the inlet–outlet hydraulic-pressure-head-drop boundary condition to the field. We employ Shannon entropy along the lines of Zehe et al. (2021) to quantify the emergence of transport self-organization in the medium. In Sect. 2, we review the basic methodology of our reactive setup approach, followed by the discussion of the concept of self-organization in the context of transport in a porous medium, as well as its quantification and relation to hydraulic power in Sect. 3. We present and discuss the obtained results in Sect. 4.

## 2 Reaction, flow, and transport modeling

### 2.1 Chemical reaction model

The key component in dissolution–precipitation reactions is the stoichiometric equilibrium between the reactants and the products. In the system under investigation, the simulated reactive-transport scenario is that of an injected acid compound (low-pH water), establishing local equilibrium with the resident fluid (high-pH water) and the porous medium, composed of calcite mineral. This fairly common setting in the field of geosciences constitutes the specific case of Edery et al. (2011), where de-dolomitization is also included in the chemical model (see also Singurindy and Berkowitz, 2003; Al-Khulaifi et al., 2017). When the low-pH water (pH level of 3.5) enters the calcite porous medium, initially saturated and in chemical equilibrium with the resident fluid (pH level of 8), dissolution of calcite occurs, accompanied by the production of calcium and carbonic acid, as represented by the following equations:

$$CaCO_{3(s)} \leftrightarrow Ca^{2+} + CO_3^{2-}, \tag{R1}$$

$$CO_3^{2-} + 2H^+ \leftrightarrow H_2CO_3. \tag{R2}$$

Here, the carbonate ion $CO_3^{2-}$, which appears due to dissolution of $CaCO_3$ in water, recombines with two hydrogen ions $H^+$ to produce carbonic acid. The deprotonation reaction (R2) represents the sum of the two acid equilibria of the carbonic acid (Manahan, 2000). The fluid pH level is assumed to be bounded by that of the resident fluid. The reaction is assumed to be fully reversible; therefore, the opposite precipitation reaction is also possible, given that the pH conditions are favorable. The injected fluid is a source of both $Ca^{2+}$ and $H^+$. We assume an abundance of $Ca^{2+}$ in the fluid, thus making the reaction (R1) non-rate-limiting; therefore, the rate-limiting reaction is the reaction (R2), which is controlled by the available hydrogen ions $H^+$. This is consistent with experimental observations (Singurindy and Berkowitz, 2003). Combining both equations along the lines of Edery et al. (2021), we obtain the following simplified equation:

$$A \leftrightarrow B, \tag{R3}$$

where $A$ and $B$ represent $2H^+$ and $H_2CO_3$, respectively. The direction of the reaction is governed by the deviation from the chemical equilibrium, as defined by the current concentrations of $H^+$ and $H_2CO_3$ in the localized area of reaction. In the presence of $H^+$ concentration above the equilibrium value, dissolution will occur and vice versa. While pH levels control the dissolution–precipitation reaction, the hydrogen ion concentration that establishes the pH is controlled by the transport processes of the invading fluid.

### 2.2 Flow and transport simulation

To investigate the dynamic coupling between the transport and reactive processes in the porous medium, we consider a two-dimensional field of dimensions $L_x \times L_y = 12\,\text{cm} \times 30\,\text{cm}$, made of calcite mineral. Here, $L_x$ and $L_y$ are the domain dimensions in the directions parallel and transverse to the flow, respectively. The field is assumed to be fully saturated with a resident high-pH water and is initially in chemical equilibrium. Low-pH water is injected at the inlet boundary, causing a reaction inside the porous medium

as it advances. The field is discretized into $N_x \times N_y$ computational cells, where $N_x = 150$ and $N_y = 60$ are the number of cells in the directions parallel and transverse to the flow, respectively, with each cell having dimensions of $\Delta x, \Delta y = 0.2\,\text{cm}$. The field is initially homogeneous in terms of hydraulic conductivity $K$ and porosity $\theta$, which have initial values of $K_0 = 10.9869\,\text{cm}\,\text{min}^{-1}$ and $\theta_0 = 0.4$, respectively. A hydraulic-head-drop boundary condition $\Delta h_{\text{BC}}$ is applied between the field's inlet and outlet boundaries, while the upper- and lower-field boundaries are ideal (reflective) walls. The flow within the field, subject to the boundary conditions as described above, is governed by the continuity and the Darcy equations (Eqs. 1 and 2) and is solved in terms of the hydraulic-pressure head with the help of a finite-element-method computer code (Guadagnini and Neuman, 1999). The obtained hydraulic-head distribution is then converted into flow velocity at each computational cell.

$$\nabla \cdot \boldsymbol{q}(\boldsymbol{x}) = 0 \tag{1}$$

$$\boldsymbol{q}(\boldsymbol{x}) = -K\nabla h(\boldsymbol{x}) \tag{2}$$

Here $\boldsymbol{q}$ and $h$ are the Darcy flux and the hydraulic-head distributions over the field, and $K$ is the hydraulic conductivity, all functions of the field spatial coordinate $\boldsymbol{x}$. Here and throughout the paper, bold font is used to specify vector fields as opposed to scalar fields, which are written in the regular font.

The solute transport across the field is simulated using a Lagrangian particle-tracking approach (Le Borgne et al., 2008). The invading fluid is represented by the reactive $A$ particles injected into the porous medium. A total number of $N_{\text{tot}} = 5 \times 10^5$ $A$ particles, representing the available pore volume in the field at pH 3.5, is injected per pore volume time at the inlet boundary at a constant rate that is proportional to the mean flow velocity in the initial homogeneous field so that, in every computational time step $\Delta t$, the amount of injected particles is $v_0 \cdot (\Delta t/L_x) \cdot N_{\text{tot}}$, where $v_0$ is the mean flow velocity magnitude within the initial homogeneous field (assuming that the change in the mean conductivity value is minor over the simulation time). The mean velocity $v_0$ is calculated from the applied pressure head drop over the field $\Delta h_{\text{BC}}$ using Darcy's law: $v_0 = K_0/\theta_0 \cdot \Delta h_{\text{BC}}/L_x$. The particles injected at each time step are flux weighted according to the conductivity distribution of the inlet cells. The particle injection rate is such that all of the available pore volume in the field is sampled by the particles by the time all $N_{\text{tot}}$ of them have been injected. The time of this occurrence will be referred to as pore volume time $T_{\text{pv}} = L_x/v_0$ throughout the paper.

The injected $A$ particles are advanced in the field using the Langevin equation, which combines deterministic (advective) and stochastic (diffusive) contributions (Risken, 1996). The position $\boldsymbol{d}(t)$ of a particle due to the combined effect of advective and diffusive transport mechanisms is described by Eq. (3), where $\boldsymbol{v}$ is the flow velocity field, $t$ is the computational time, $D$ is the diffusion coefficient of the in-

vading fluid in the porous medium, and $\boldsymbol{\xi}(t)$ is a vectorial Gaussian random variable characterized by $\langle\boldsymbol{\xi}(t)\rangle = \boldsymbol{0}$ and $\langle\xi_i(t)\xi_j(t')\rangle = \delta_{ij}\delta(t - t')$.

$$\frac{\mathrm{d}\boldsymbol{d}(t)}{\mathrm{d}t} = \boldsymbol{v}[\boldsymbol{d}(t)] + \sqrt{2D}\boldsymbol{\xi}(t) \tag{3}$$

An important property of the Langevin Eq. (3) is its equivalence to the well-known advection–diffusion equation (ADE) (Risken, 1996; Perez et al., 2019). To be used in the context of a numerical simulation, the equation is discretized using the simple Euler–Maruyama method (Kloeden, 1992) as follows:

$$\boldsymbol{d}_{k+1} = \boldsymbol{v}_k \cdot \Delta t + \boldsymbol{d}_D, \tag{4}$$

where $\boldsymbol{d}_{k+1} = \boldsymbol{d}(t_{k+1})$ is the particle location at the current computational time step $t_{k+1} = \Delta t \cdot (k+1)$, and $\boldsymbol{v}_k = \boldsymbol{v}[\boldsymbol{d}(t_k)]$ is the flow velocity at the particle location at the previous computational time step $t_k = \Delta t \cdot k$ (here, the index $k$ is not to be confused with the hydraulic conductivity $K$). The diffusive contribution is given by $\boldsymbol{d}_D = \sqrt{2D\delta t} \cdot \boldsymbol{N}(0, 1)$, where $D = 1 \times 10^{-5}\,\text{cm}^2\,\text{min}^{-1}$ is the representative value of the diffusion coefficient (Domenico and Schwartz, 1997), $\boldsymbol{N}(0, 1)$ is a standard normally distributed random variable, and $\delta t = \delta s/v$ is the time it takes a particle to move the fixed distance $\delta s = \Delta x/10$ while traveling with the flow at a speed of magnitude $v = |\boldsymbol{v}|$. The obtained diffusive contribution has no spatial preference since its spatial direction is represented by a uniformly distributed random variable between $[0, 2\pi]$. The particle movement in the duration of the computational time step $\Delta t$ consists of a series of jumps of a constant magnitude $\delta s$, to which the diffusive contribution is added. This series of jumps continues until their cumulative time $\sum_i \delta t_i$, where $i$ is the jump index in the series, reaches $\Delta t$.

At first, no $B$ particles exist in the field. After all newly injected $A$ particles have been advanced to their current locations, they are allowed to react according to an algorithm described in Sect. 2.3. Following reaction, as described by Reaction (R3), $B$ particles appear in the field, and the computational cell values of porosity and hydraulic conductivity in the area of reaction are updated correspondingly, thus creating an interaction between the reactive and the transport processes. Beginning the next computational step $\Delta t$, another set of $A$ particles is injected into the field, employing flux weighting in accordance with the reaction-modified conductivity distribution of the inlet cells. Then, all available particles in the field (both $A$ and $B$) are advanced again, and reaction occurs. Note that the diffusive contribution to particle displacement $\boldsymbol{d}_D$ allows for the mixing effect in the field due to its stochastic nature. Following the changes in the hydraulic-conductivity field due to the reactive process, the hydraulic-pressure-head field and the corresponding flow velocity field are updated at constant time intervals of $10\Delta t$ (in order to reduce constraints associated with computational costs), while the reaction is allowed to occur every $\Delta t$. The simulation continues until the pore volume time

$T_{pv}$ is reached, while the $A$ particle injection rate remains constant throughout the simulation. Hydraulic conductivity, pressure head, porosity, and velocity fields, as well as the amount of dissolution–precipitation reactions that took place in each computational cell, are recorded at constant time intervals of $10\Delta t$ to be analyzed in the post-processing stage. The particle transport algorithm and the chemical reaction model described above constitute two important aspects of the kinetic reaction mechanism as implemented in the current study.

The transport part of the model has been validated against the well-known case of one-dimensional instantaneous injection in a homogeneous medium, for which an analytical solution exists (Kreft and Zuber, 1978). See Sect. S3 in the Supplement for details.

## 2.3 Kinetic reaction mechanism

In the system under investigation, the kinetic reactive process operates according to an algorithm developed to mimic the actual chemical dissolution–precipitation reactions that take place in practice. After all existing $A$ and $B$ particles in the field have been advanced in the current time step, the reaction is allowed to occur. Since the reaction is assumed to be locally instantaneous, at each instant of computational time $\Delta t$ the reaction is allowed to proceed until local equilibrium is reached in each field cell. For that purpose, the number of currently residing particles of both kinds is assessed in each cell. Each $A$ particle is assigned a molar amount based on the assumption that the total quantity of $A$ particles injected per pore volume time $N_{\text{tot}}$, distributed evenly across the field, results in the pH level of 3.5 throughout the field, which corresponds to the pH level of the invading fluid. The following calculation is performed: a total available pore volume in the computational field $\theta_0 L_x L_y$ is multiplied by the molar density of the hydrogen ions that are injected into the field, corresponding to a pH level of 3.5. This gives $4.55 \times 10^{-5}$ mol of hydrogen ions that are required to fill the initially available pore volume to obtain a pH level of 3.5 throughout the field. To obtain the molar amount assigned to a single $A$ particle, this value is divided by $N_{\text{tot}}$. For $N_{\text{tot}} = 5 \times 10^5$ $A$ particles, as defined in the current simulation, each one obtains a parcel of $9.1 \times 10^{-11}$ mol of H$^+$ (recall that $A = 2$H$^+$). Based on Reaction (R3), each $B$ particle obtains half of this molar amount of carbonic acid. The number of $A$ particles required to obtain pH 3.5 in a particular cell is calculated by dividing $N_{\text{tot}}$ particles by the number of computational cells in the field $N_x \cdot N_y$ to obtain approximately 56 (while zero $A$ particles in a cell represents a pH level of 8, which corresponds to the pH of the resident fluid). Based on the actual number of $A$ particles in the cell, the current pH level is estimated using a linear proportion from these limiting values. Since Ca$^{2+}$ is not the rate-limiting factor, reaching local equilibrium in the cell amounts to equilibrating the carbonates. The equilibrium value of the fractional amount of H$_2$CO$_3$ in relation to the total quantity of carbonates in a cell, denoted by $\alpha_1(\text{pH})$, is calculated based on the current pH value in the cell as

$$\alpha_1(\text{pH}) = \frac{10^{-2\text{pH}}}{10^{-2\text{pH}} + 10^{-\text{pk}_{a1}} + 10^{-\text{pk}_{a2}} + 10^{-\text{pH}-\text{pk}_{a1}}}, \quad (5)$$

with the standard values $\text{pk}_{a1} = 6.35$ and $\text{pk}_{a2} = 10.33$ (Manahan, 2000) (see Supplement for Edery et al., 2011, for details). The value of $\alpha_1(\text{pH})$ is then compared to the current fractional amount of H$_2$CO$_3$ in a cell, calculated based on an assumption that the total quantity of carbonates in a cell at any time corresponds to the amount of H$^+$ that results in a pH level of 3.5. The direction of the reaction may now be determined in accordance with Reaction (R3): dissolution will occur if the fractional amount of H$_2$CO$_3$ is smaller than the equilibrium value and vice versa. A single reaction is allowed to occur, during which a single $A$ particle in a cell transforms into a $B$ particle or vice versa. After each reaction turn, the local pH level in a cell and the fractional amount of H$_2$CO$_3$ there in equilibrium $\alpha_1(\text{pH})$, as well as the fractional amount of H$_2$CO$_3$ in practice, are recalculated and the reaction process is repeated until equilibrium is reached.

After that, the cell porosity $\theta$ is updated in accordance with Eq. (6), where the subscripts $k + 1$ and $k$ denote values at the current and previous computational steps, respectively (before and after the reaction has taken place). Here, the porosity increment is actually the volume of calcite, dissolved or precipitated due to the reaction, divided by the volume of the cell $\Delta x \Delta y$ (assuming unit cell depth). According to Reactions (R1) and (R2), the change in cell volume is equal to the change in the molar amount of H$_2$CO$_3$ due to the reaction in the cell $d[\text{H}_2\text{CO}_3]$ (can be positive or negative, depending on the direction of the reaction) times the molar volume of calcite, taken to be $M_{\text{CaCO}_3} = 36.93$ cm$^3$ mol$^{-1}$ (Morse and Mackenzie, 1990). The porosity is not allowed to exceed limiting values of 0.01 and 0.99, set to avoid occurrence of nonphysical scenarios. The hydraulic conductivity $K$ is then updated employing the Kozeny–Carman relation (Eq. 7).

$$\theta_{k+1} = \theta_k + \frac{d[\text{H}_2\text{CO}_3]M_{\text{CaCO}_3}}{\Delta x \Delta y} \quad (6)$$

$$K_{k+1} = K_k \cdot \frac{\theta_{k+1}^3}{(1-\theta_{k+1})^2} \cdot \frac{(1-\theta_k)^2}{\theta_k^3} \quad (7)$$

This process is repeated for each of the cells in the field before the particles are allowed to advance again in the next computational time step. In order to overcome constraints associated with computational costs, reaction enhancement has also been considered. For this purpose, the change in the cell volume (Eq. 6) due to a certain amount of reaction that took place there is increased by several orders of magnitude. This is equivalent to accelerating the reactive process in the field, while the overall dynamics associated with the reactive process remain the same.

## 2.4 Definition of the Peclet number

The Peclet number plays an important role in our reactive system. As a usual practice, Peclet number is calculated based on an Eulerian length scale, such as the mean grain or pore diameter in the pore-scale simulation or the characteristic correlation length of the heterogeneous porous media for the case of the Darcy-scale simulation (Nguyen and Papavassiliou, 2020). The Eulerian definition of the Peclet number as the ratio of advective to diffusive transport rates gives the following well-known relation: $Pe = \tilde{v}L/D$, where $\tilde{v}$ is the mean velocity, and $L$ is the characteristic length. This definition often yields estimates that allow little physical insight into the subject of the relative contributions of the advective and diffusive processes to the transport in porous media under investigation. Moreover, in the case of reactive transport in an initially homogeneous porous medium on the Darcy scale, such as in the current study, where heterogeneity is gradually introduced in the field, correlation length can be time dependent; this calls for alternative approaches in defining the Peclet number that are capable of giving a deeper insight into the transport characteristics. In order to find a meaningful estimate for this parameter, we attempt to formulate the expression for the Peclet number based on Lagrangian length scale quantities, noting, after Nguyen and Papavassiliou (2020), that the nature of hydrodynamic dispersion is Lagrangian. For this purpose we turn to the Langevin Eq. (3), which governs the particle advancement in the simulation. Each jump made by a particle during the time period $\delta t$ consists of a constant-value advective contribution $\delta s = \Delta x/10$ and a diffusive contribution $\sqrt{2D\delta t} \cdot N(0, 1)$ (see Sect. 2.2). Only the magnitude of the diffusive contribution is of importance; therefore, we will consider its absolute value $\sqrt{2D\delta t} \cdot |N(0, 1)|$. The random variable $|N(0, 1)|$ is half-normally distributed, with an expected value of $\sqrt{\left(\frac{2}{\pi}\right)}$ (Leone et al., 1961). Therefore, we are able to formulate the expression for the Peclet number directly from its definition as the ratio of advective to diffusive transport rates, given by $Pe = \delta s/\sqrt{4/\pi D \delta t}$. The time duration of a single jump is determined as $\delta t = \delta s/v$, where $v$ is the magnitude of the fluid velocity at the current particle location. Assuming that the change in the mean conductivity value is minor over the simulation time, we can replace $v$ with the initial flow velocity $v_0 = K_0/\theta_0 \cdot \Delta h/L$. Thus, we finally obtain the expression for the Lagrangian transport Peclet number in our simulation:

$$Pe = \sqrt{\frac{\pi \delta s K_0 \Delta h_{BC}}{4DL_x\theta_0}}. \tag{8}$$

The above definition provides a direct estimate of the relative contributions of the advective and diffusive processes to the transport in porous media. This approach may prove useful for Lagrangian particle-tracking methods where the advective and diffusive transport contributions are defined explicitly.

To investigate the influence of the Peclet number of the transport on the evolution of the reactive process and the emergence of transport self-organization in the field, we simulate a number of reactive-transport scenarios for different values of the Peclet number. This is achieved by applying different values of the inlet–outlet hydraulic-head-drop boundary condition $\Delta h_{BC}$. The computational time step $\Delta t$ is correlated with $\Delta h_{BC}$ in such a way that, in each simulated reactive-transport scenario, corresponding to a specific value of applied $\Delta h_{BC}$, an equal number of 1800 time steps $\Delta t$ and, correspondingly, reaction events occur per pore volume time $T_{pv}$.

## 3 Identifying and quantifying self-organization

### 3.1 Emergence of transport self-organization in reactive transport in an initially homogeneous porous media

The computational setting described in the previous section mimics the dynamics of a coupled dissolution–precipitation reactive process in a calcite porous medium, leading to the emergence of heterogeneity in an initially homogeneous field. Previous studies have shown that self-organization of the solute transport in the field is expected to emerge in such a situation in the form of preferential flow paths that lead to solute concentration gradients in the direction transverse to flow (Zehe et al., 2021), yet the details of this self-organization emergence and evolution are critical to understanding the large-scale dynamics of the coupled reactive process in the field. To analyze the emergence of self-organization in our computational field, we consider snapshots of the field in terms of the hydraulic-conductivity distribution, taken at different computational times as the reactive process in the field evolves. We consider each snapshot to be an open thermodynamic system and perform a non-reactive tracer test by injecting non-reactive solute at the field's inlet. Along the lines of Sect. 1.2, we argue that organized states, characterized by reduced entropy, can emerge in an open system, driven away from equilibrium due to the exchange of energy or matter with surroundings. Such a system may persist in a stationary non-equilibrium state. Since, according to the second law, overall entropy cannot decrease, in such a case entropy must be exported from the system outside, leading to an increase in the entropy of its surroundings.

Section S1 in the Supplement presents a simple heat transfer example that illustrates how an open thermodynamic system can be maintained in a stationary non-equilibrium state through an inflow of energy. Following the same concept, one can generalize this finding for the system under investigation, represented by a snapshot of the reactive field at a specific computational time, where the non-reactive solute

transport self-organization in the field in terms of emergence of preferential flow paths is the outcome of the coupled reactive process that introduces heterogeneity in an initially homogeneous field. In our system, energy influx occurs in the form of hydraulic power, supplied to the flow to overcome the hydraulic resistance of the field by the applied hydraulic-pressure-head-drop boundary condition $\Delta h_{BC}$. Such a system may persist in a non-equilibrium stationary state, characterized by the lowered entropy of the transversal distribution of solute concentration, with the hydraulic power, supplied to the flow, acting against depletion of transversal concentration gradients. For our system, an equilibrium state corresponds to the state of perfect mixing in the field, where no such gradients exist (Zehe et al., 2021).

To identify the driving mechanism that leads to an emergence of heterogeneity in an initially homogeneous porous medium followed by self-organization of the solute transport in the field, let us consider two limiting cases related to the nature of the transport mechanism in the field, as applicable to our reactive-transport setting, described in Sect. 2. First, we consider a case where no advection is present (the advective velocity $v = 0$). In this scenario, reactive $A$ particles advance within the field due to diffusive action only. Because of the stochastic nature of the diffusive process that lacks spatial preference, and since the diffusive properties of the reactant and the product particles are identical (also, in our model, diffusion is independent of porosity), we expect that the dissolution and precipitation reactions will occur uniformly in space; thus, the conductivity field will remain approximately homogeneous in the direction transverse to flow. Therefore, we suggest that, in the absence of advection, no mechanism exists in our reactive setup to create heterogeneity in the field, and no transport self-organization will take place in that case. In the opposite limiting case where no diffusion is present (the diffusion coefficient $D = 0$), a dissolution reaction will first take place uniformly at the inlet of the initially homogeneous field as the reactive $A$ particles enter the field. The resulting carbonic acid particles, in accordance with Reaction (R3), will be swept downstream by the advective flow to cause a precipitation reaction uniformly along the direction transverse to flow at some distance downstream. This pattern of alternating dissolution and precipitation areas at identical intervals in the downstream direction will repeat itself, with the resulting pattern being reminiscent of a precipitation-banding phenomenon (Singurindy and Berkowitz, 2003). Again, in this case, no heterogeneity in the direction transverse to flow is expected to emerge; thus, we suggest that no transport self-organization will occur as well.

Following the above discussion, we argue that, for the transport self-organization to emerge in an initially homogeneous field undergoing a dissolution–precipitation reactive process as defined in our reactive setup, it is necessary for the transport mechanism to include both diffusive and advective contributions. Here, the stochastic diffusion leads to local concentration variations in $H^+$ that, in turn, create lo-

cal variations in hydraulic conductivity, while the advection follows these conductivity variations, funneling the flow towards the higher-conductivity areas and further increasing their conductivity due to the enhanced dissolution by means of the funneled $A$ particles. This creates positive coupling between the reactive and transport processes. Here, an analogy can be made to the Turing's morphogenesis model (see Sect. 1.2), where the homogenizing action of the diffusion is counteracted by the fact that the diffusion coefficients of the reacting substances differ, which is a prerequisite for the appearance of self-organizing patterns. We suggest that the driving force for transport self-organization in the model under investigation is the ratio between the diffusive and advective transport rates or the reciprocal of the Peclet number. Therefore, we expect to obtain an increase in transport self-organization in the field with a decrease in the Peclet number of the flow.

In an attempt to derive an analytical justification for this, let us relate the heterogeneity, emerging in an initially homogeneous field due to the coupled reactive-transport process, to the global reaction rate in the field. The reasoning behind this is that, in the initially homogeneous field, reaction that occurs in the field is the driving force behind the emergence of heterogeneity there; heterogeneity, in turn, is responsible for the emergence of transport self-organization in the field. While some of the reaction events that occur may cancel each other out in terms of dissipating or precipitating some of the calcite in a computational cell, the remaining part of reaction is useful in creating heterogeneity. Thus, we may speculate that the reaction rate and the emergence of heterogeneity in the field are directly related. Consider, for simplicity, a 1D advection–diffusion reaction equation (ADRE):

$$\frac{\partial c}{\partial t} = -v \frac{\partial c}{\partial x} + D \frac{\partial^2 c}{\partial x^2} - G/\theta, \tag{9}$$

where $c$ is the solute concentration, $v$ is the advective velocity, $D$ is the diffusion coefficient, $G$ is the reactive term, and $\theta$ is the porosity of the medium. Let us consider a simple scenario of an adsorption–desorption reaction, for which parallels can be drawn to the dissolution–precipitation scenario in our study (Raveh-Rubin et al., 2015). The simplest case would be to employ a linear case of the Freundlich isotherm model, where the adsorption is directly proportional to the concentration (see Berkowitz et al., 2014, and Bear and Cheng, 2010)

$$: A = k_d c, \tag{10}$$

where $A$ is the amount of adsorption, and $k_d$ is the distribution (sorption) coefficient. For desorption, the sign of the reaction term should be reversed. The reactive term $G$ obtains the following shape:

$$G = \rho_b \frac{\partial A}{\partial t} = k_d \rho_b \frac{\partial c}{\partial t}, \tag{11}$$

where $\rho_b$ is the dry bulk density of the porous medium. After some manipulation, the ADRE for the adsorption case may be written as

$$R \frac{\partial c}{\partial t} = -v \frac{\partial c}{\partial x} + D \frac{\partial^2 c}{\partial x^2}, \tag{12}$$

where $R = 1 + k_d \frac{\rho_b}{\theta}$ is the retardation factor. To obtain the non-dimensionalized version of ADRE, we normalize all variables as follows: $\tilde{c} = c/c_{\text{ref}}$, where $c_{\text{ref}}$ is the reference value for the solute concentration; $\tilde{x} = x/L_x$; and $\tilde{v} = v/v_0$. The temporal coordinate is normalized by the pore volume time $T_{\text{pv}}$ to obtain $\tilde{t} = tv_0/L_x$. Thus, the resulting non-dimensionalized version of ADRE is

$$R \frac{\partial \tilde{c}}{\partial \tilde{t}} = -\tilde{v} \frac{\partial \tilde{c}}{\partial \tilde{x}} + \frac{1}{Pe} \frac{\partial^2 \tilde{c}}{\partial \tilde{x}^2}, \tag{13}$$

where $Pe = L_x v_0/D$ is the classical definition of the Peclet number. Recalling the linear relation between the adsorption and the concentration (Eq. 10), we may write for the non-dimensional adsorption rate

$$\frac{\partial \tilde{A}}{\partial \tilde{t}} = \frac{k_d c_{\text{ref}}}{R A_{\text{ref}}} \left( -\tilde{v} \frac{\partial \tilde{c}}{\partial \tilde{x}} + \frac{1}{Pe} \frac{\partial^2 \tilde{c}}{\partial \tilde{x}^2} \right), \tag{14}$$

where $\tilde{A} = A/A_{\text{ref}}$, with $A_{\text{ref}}$ being the reference value for the adsorption amount. To obtain the global reaction rate in the field, Eq. (14) should be integrated over the domain. For the case when the advective transport contribution is small relative to the diffusive one $\left( -\tilde{v} \frac{\partial \tilde{c}}{\partial \tilde{x}} \ll \frac{1}{Pe} \frac{\partial^2 \tilde{c}}{\partial \tilde{x}^2} \right)$, this result suggests the dependence of the global reaction rate on the reciprocal of Peclet for our simple model. Recalling an assumption made earlier about the heterogeneity and, thus, the ensuing transport self-organization in the field being directly related to the global reaction rate, we arrive at a conclusion that both these parameters depend on the reciprocal of the Peclet number for diffusion-dominated flows.

## 3.2 Self-organization quantization employing Shannon entropy

Having obtained a qualitative understanding of the subject of self-organization in the context of reactive transport of the porous medium, we shall now seek a way to characterize this phenomenon quantitatively using the concept of Shannon entropy, also referred to as information entropy. Shannon entropy was introduced originally in the field of communication theory, whose fundamental problem is formulated as "reproducing at one point either exactly or approximately a message, selected at another point" (Shannon, 1948). Viewing communication as a statistical process, Shannon employed an entropy-like parameter to provide a measure for the amount of transmitted information carried by a certain sequence of symbols (a message). Section S2 in the Supplement contains a short account of the Shannon entropy,

while, here, we only reprint the main result:

$$S = -\Sigma_i \, p_i \log_2 p_i, \tag{15}$$

where $S$ is the Shannon entropy per symbol of the message, and $p_i = n_i/n$, $i = 1 \dots N_s$ are the relative occurrence frequencies of $N_s$ different symbols that constitute the message, calculated as a ratio of the number of occurrences of a specific symbol $i$ in the message $n_i$ to the total message length $n$.

The definition of information entropy, given by Shannon, is equivalent to physical entropy in statistical mechanics as defined by Gibbs, where the logarithm in Eq. (15) is in relation to the base of $e$, and the sum is multiplied by the Boltzmann constant (Ben-Naim, 2008). The statistical definition of physical entropy characterizes the number of possible microstates of a system that are consistent with its macroscopic thermodynamic properties which constitute the macrostate of the system. Thus, in the case of a gas consisting of a large number of molecules in a container, a microstate of the system consists of the position and momentum of each molecule as they move within the container, colliding with other molecules and container walls. A multitude of such microstates correspond to a single macroscopic state of the system, defined by its pressure and temperature. The parameter $p_i$ in this case corresponds to a probability that a microstate $i$ occurs during the system's fluctuations. According to the second law of thermodynamics, entropy of a system reaches its maximum value at equilibrium, where gradients of thermodynamic parameters are depleted by dissipative forces and where the measure of order in the system is at its minimum. In this case, each microstate is equally likely, and $p_i$ is simply the inverse of the total number of microstates (Kondepudi and Prigogine, 1998).

To characterize the emergence and development of transport self-organization in an initially homogeneous field as the dissolution–precipitation reactive processes in the field evolve, we adopt a straightforward use of the Shannon entropy, in a similar vein as Zehe et al. (2021), where it was employed in the context of characterizing self-organization in heterogeneous non-reactive groundwater systems. In order to place all results on equal footing, we perform additional computational non-reactive tracer tests on the snapshots of the hydraulic-conductivity field at different simulation times, as described in Sect. 3.1. The non-reactive particle tracer algorithm, employed for this purpose, consists of the flow and particle transport algorithm described in Sect. 2.2 but without the kinetic-reaction part. A total number of $N_{\text{tot}}^{\text{NR}} = 1 \times 10^5$ particles, which represent the non-reactive tracer solute (not to be confused with the number of reactive particles $N_{\text{tot}}$ injected per $T_{\text{pv}}$ in the reactive algorithm), are injected at the inlet of the computational field, subject to the identical hydraulic-pressure-head-drop boundary condition of $\Delta h_{\text{BC}} = 100 \, \text{cm}$, and are allowed to advance within the field subject to the laws described in Sect. 2.2 until all of them reach the outlet boundary. The relatively large value of $\Delta h_{\text{BC}}$ is chosen in order to eliminate the diffusive transport

contribution, which tends to smooth out concentration gradients due to its stochastic character. The injected particles are flux weighted according to the conductivity distribution of the inlet cells. The field density matrix, obtained by counting the total number of particle visitations in each computational cell, is saved for each field snapshot. From this matrix, the solute concentration distribution of the non-reactive tracer across the field is obtained. We recall the proposition put forth by Berkowitz and Zehe (2020) that solute transport self-organization corresponds to an emergence of solute transport gradients in the field in the direction transverse to the direction of the flow. To quantify the entropy of transport self-organization in the snapshots of the hydraulic-conductivity field, we rewrite Eq. (15) as follows:

$$S(x) = -\Sigma_i p_i \log_2 p_i, \tag{16}$$

where $S(x)$ is the Shannon entropy of the transversal solute concentration at a given axial coordinate $x$, and $p_i = N_i/N_{\text{tot}}^{\text{NR}}$, $i = 1 \dots N_y$ is the relative solute concentration distribution at $x$ in the direction transverse to flow (here, $N_i$ is the number of particles that have passed through the $i$th cell at $x$).

### 3.3 The relation between self-organization, heterogeneity, and hydraulic power

Following the thermodynamic framework, as applied to the system under investigation, energy must be invested in the heterogeneous field to maintain an ordered state of solute transport there. This energy comes in the form of hydraulic power that enables the flow to overcome the hydraulic resistance of the porous medium under the applied hydraulic-pressure-head-drop boundary condition $\Delta h_{\text{BC}}$. The total hydraulic power dissipated by the flow while overcoming the hydraulic resistance of the field is an extensive parameter and is given by a cell-wise summation:

$$P_{\text{total}} = -\rho_f g \sum_{i,j} \Delta h_{ij} \cdot Q_{ij}, \tag{17}$$

where $i$ and $j$ are the row and the column indices of the $(i, j)$th cell in the field, respectively; $\Delta h_{ij}$ and $Q_{ij}$ are the hydraulic-head drop and the volumetric flow rate through the $(i, j)$th cell; $\rho_f$ is the fluid density; and $g$ is the gravitational acceleration.

In an attempt to better understand the relation between the degree of heterogeneity of the porous medium and the hydraulic power required for the fluid to overcome the resistance of the medium under the inlet–outlet hydraulic-head-drop boundary condition $\Delta h_{\text{BC}}$, we consider a simplified model of the field as a series of conductive channels connected in parallel. These channels have no interaction transversally, and the deviations of their shape from the shortest path between the inlet and the outlet (a straight line) is not significant. This model corresponds to a moderately heterogeneous field that consists of a number of preferential

flow paths where most of the solute transport (and, thus, most of the reaction) occurs. Assuming that the heterogeneity introduced into the field due to the coupled reactive-transport process is minor, the shape of the obtained preferential flow paths should indeed not stray too far from a straight line. Each conductive channel is represented by a collection of cells along the preferential flow path in the field. The equivalent hydraulic conductivity of the $i$th single channel is

$$K_i^{\text{EQ}} = L_i / \underset{j}{\Sigma} \Delta l_j \cdot K_{ij}^{-1}, \tag{18}$$

where the index $j$ runs along the cells that constitute the $i$th channel, $L_i$ is the total length of the channel, and $K_{ij}$ and $\Delta l_j$ are the hydraulic conductivity and the length of the preferential path segment inside the $j$th cell of the $i$th channel (do not confuse with the row and column indices $i$, $j$ as defined earlier. For the most general case of a curvilinear flow path the $i$, $j$ indices may not conform to the row and column indices). The equivalent hydraulic conductivity of the whole field, viewed as a system of conductive channels, connected in parallel, is

$$K_{\text{field}}^{\text{EQ}} = \sum_i K_i^{\text{EQ}} \cdot \frac{\Delta y}{L_y} = \frac{\Delta y}{L_y} \sum_i \frac{L_i}{\Sigma_j \Delta l_j \cdot K_{ij}^{-1}}. \tag{19}$$

Under the assumption of parallel channels with no interaction taking place between them transversally, the total hydraulic power dissipated in the field is given by

$$P_{\text{total}}^{\text{EQ}} = -\rho g \Delta h_{\text{BC}} \cdot Q_{\text{total}} = \rho g \left( \frac{\Delta h_{\text{BC}}}{L_x} \right)^2 L_x L_y K_{\text{field}}^{\text{EQ}}, \tag{20}$$

where $Q_{\text{total}}$ is the total volumetric flow rate through the field. To relate the increase in hydraulic power with an emergence of heterogeneity in the field, which leads to the increasing transport self-organization, consider again the relations of Eqs. (19) and (20) given above. The total hydraulic power in the field, calculated using the parallel channels assumption $P_{\text{total}}^{\text{EQ}}$, is related to the hydraulic-conductivity distribution by virtue of the equivalent hydraulic conductivity of the field (Eq. 19), which can be rewritten as

$$K_{\text{field}}^{\text{EQ}} = \frac{\Delta y}{L_y} \sum_i \frac{1}{R_i}, \tag{21}$$

where $R_i = (1/L_i) \sum_j \Delta l_j \cdot K_{ij}^{-1}$ is the reciprocal of the equivalent hydraulic conductivity of the $i$th channel $K_i^{\text{EQ}}$ or its hydraulic resistivity. Recall the well-known feature of the harmonic mean of some population: if the population is subjected to a mean-preserving spread (that is, its variance is increased while the mean is kept at a constant value) then its harmonic mean always decreases (Mitchell, 2004). Since Eq. (21) can be viewed as a reciprocal of the harmonic mean of the population of hydraulic channel resistivities $R$, we expect that $K_{\text{field}}^{\text{EQ}}$ will increase with an increase in the variance

of $R$, assuming the change in the mean hydraulic conductivity of the field is minor. Moreover, since the variance of $R$ depends directly on the variance of the hydraulic-conductivity distribution $K$ in the field, it is reasonable to expect that $K_{\text{field}}^{\text{EQ}}$ will increase with an increase in the variance of $K$, which signifies the emergence of heterogeneity in the initially homogeneous field. Thus, an increase in heterogeneity of an initially homogeneous porous medium due to the coupled reactive process requires an increase in the hydraulic power, supplied to the flow to overcome the hydraulic resistance of the medium under the maintained hydraulic-head-drop boundary condition $\Delta h_{\text{BC}}$ and applied to the field. Within the adopted thermodynamic framework, this increase in power allows us to increase the spatial self-organization of the solute transport in the field. To conclude, the increase in the hydraulic power, dissipated in the field, and the emergence of transport self-organization are both the result of an increase in the heterogeneity of the field. The latter, in turn, can be viewed as a consequence of the energy invested in the field by the dissolution–precipitation reactive process.

We emphasize that the presented study does not intend to construct a complete thermodynamic formalism for the problem under investigation, such as the one that is presented in Hansen et al. (2018) and Hansen et al. (2023). The thermodynamic framework presented in the current study aims to provide qualitative dependencies and/or trends between the parameters of interest, such as the entropy of the transport self-organization, the hydraulic power dissipated in the field, etc. It is our intent to arrive at a more complete thermodynamic formalism for the reactive flow in porous medium in the course of the research work.

## 4 Results and discussion

Based on the reactive-transport algorithm described in Sect. 2, we analyze the evolution of the reactive process in an initially homogeneous reactive field and demonstrate that the parameters that characterize this evolution, such as global reaction rate, mean value, and variance of the hydraulic-conductivity distribution, indeed depend on the reciprocal of the Peclet number, as suggested by the analytic result derived in Sect. 3.1. We then investigate the emergence and evolution of transport self-organization in the field along with the advancement of the reactive process and showcase that transport self-organization correlates with the reciprocal of the Peclet number as well. To investigate these relations, we simulate a number of reactive-transport scenarios for different values of the ratio of advective to diffusive transport rates, characterized by the Peclet number and calculated as shown in Sect. 2.4. This is achieved by applying different values of the inlet–outlet hydraulic-pressure head boundary condition $\Delta h$, as described therein. Thus, for $\Delta h = 1 \times 10^{-4}, 1 \times 10^{-3}, 1 \times 10^{-2}$ and $1 \times 10^{-1}$ [cm], we obtain the Peclet number as approximately $Pe = 0.38, 1.2,$

3.8, and 12.0. A reaction enhancement by a factor of $5 \times 10^2$ is employed (see Sect. 2.3). We follow this series of analyses by quantifying the evolution of transport self organization in time as described in Sect. 3.2, showing that transport self-organization in the field increases with a decrease in the Peclet number, as expressed by the reduction in the Shannon entropy. The self-organization of the breakthrough curve exhibits the opposite tendencies, which are observed from the perspective of a thermodynamic analogy. The hydraulic power, required to maintain the driving head pressure drop boundary condition between the inlet and outlet, increases with the increasing variance of the hydraulic-conductivity distribution in the field, as suggested by the simple parallel-channels model developed in Sect. 3.3 (along with the contribution of the mean value of conductivity). This increase in power results in an increase in the transport self-organization in the field.

### 4.1 Evolution of the reactive process in the field

We begin by examining the evolution of the reactive process, as depicted by the snapshots of the relative hydraulic conductivity $K - K_0$, of the field at different normalized times $\tilde{t} = t/T_{\text{pv}}$, shown in Fig. 1a–c (obtained from a realization of the reactive process for $Pe = 0.38$). The normalized time $\tilde{t}$ is defined in such a way that, for $\tilde{t} = 1.0$, the reactive front has permeated through the whole medium. As the reactive particles advance and react in the initially homogeneous field, heterogeneity is introduced into the field in the form of local dissolution and precipitation areas, signified by positive and negative values of $K - K_0$, respectively. This heterogeneity advances downstream alongside the reactive particles, with the local dissolution–precipitation areas intensifying with time as the reactive process in the field develops, creating a clearly distinguishable reaction front. Downstream of the front, the field remains homogeneous as the reactive process has not arrived there yet. A dissolution area is located in the immediate vicinity of the inlet, where the chemical equilibrium is tilted towards dissolution; this correlates with experimental and simulation observations (Poonoosamy et al., 2020; Deng et al., 2022). The correlation length of these field snapshots was found to be of the order of the size of the computational cell; therefore, no larger structures in the hydraulic-conductivity distribution are observed.

The influence of the Peclet number on the evolution of the reactive process in the field is characterized in Fig. 2a–d. Figure 2a presents the global reaction rate $\dot{R}$, which was calculated as the mean value of the absolute rate of change of hydraulic conductivity with dimensionless time $|\mathrm{d}K/\mathrm{d}\tilde{t}|$ over all computational cells as a function of dimensionless time $\tilde{t}$. This parameter takes into account the overall number of "useful" reaction events that take place per unit time, including both dissolution and precipitation. Here, the term useful is applied to reaction events that did not cancel each other but served to alter the hydraulic-conductivity distribution in the

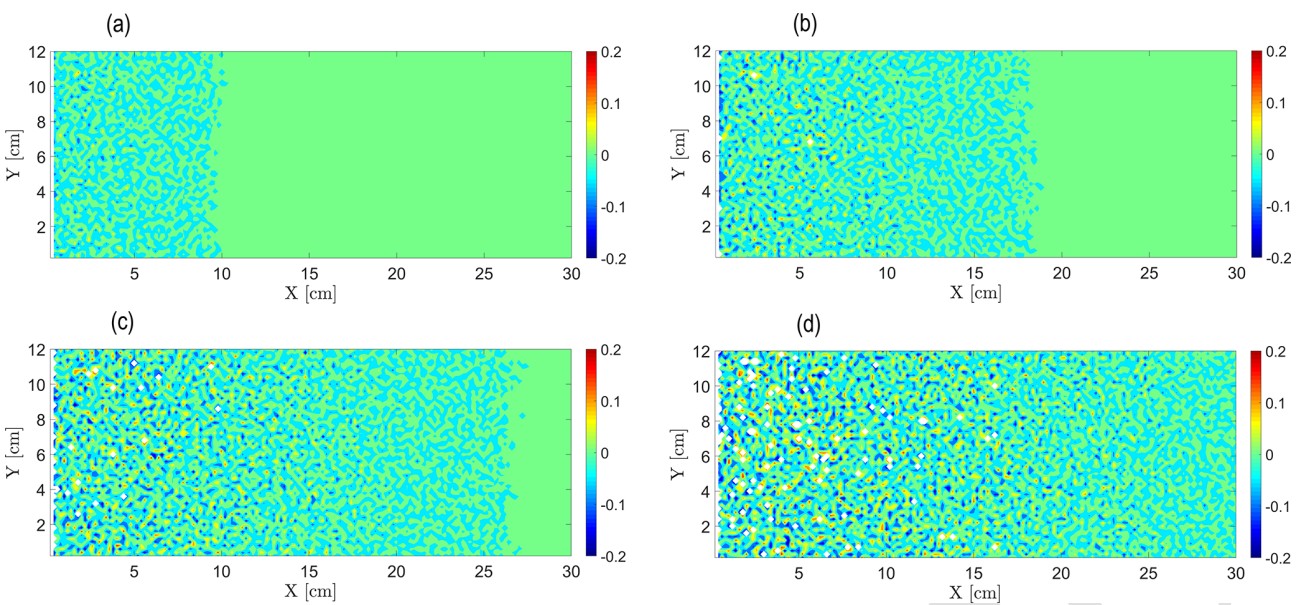

**Figure 1.** Evolution of the relative hydraulic conductivity $K - K_0$ [cm min$^{-1}$] field over time for $Pe = 0.38$: **(a)** $\tilde{t} = 0.25$, **(b)** $\tilde{t} = 0.5$, **(c)** $\tilde{t} = 0.75$, and **(d)** $\tilde{t} = 1.0$.

field. The calculation was performed excluding the area of 10 computational cells in the immediate vicinity of the inlet so as to focus on the evolution of the reactive process within the field, excluding the inlet conductivity alteration, where the reactive process is tilted towards dissolution; this effect is of a very localized nature and, thus, was ignored. We observe that the global reaction rate increases with dimensionless time in a linear fashion as the reactive particles sample more of the field's territory at an approximately constant mean velocity, which allows broader opportunities for reaction. Beginning from $\tilde{t} = 1$, $\dot{R}$ becomes approximately constant as the reactive particles have sampled an entire field at that point. The influence of the Peclet number is clearly exhibited by the fact that the global reaction rate increases with the reciprocal of the Peclet number, showing that the diffusive transport mechanism, responsible for mixing in the field, enhances the reactive process. This can be explained by the fact that diffusion, being stochastic in nature, also allows particles to sample regions in the transverse direction, away from the path suggested by the advection mechanism, which allows a better chance for reaction. This result further confirms the findings of Nissan and Berkowitz (2019) on bimolecular reaction, where an increase in the reactant production rate with a decrease in Peclet number was reported due to an increase in the spatial spread of the transported species; this trend was also reported in Edery et al. (2021).

To verify the dependency of the global reaction rate on the reciprocal of the Peclet number, as suggested by the theoretical result obtained in Sect. 3.1, the scalability of the curves in Fig. 2a with the Peclet number was investigated. Assuming separation of variables in $\dot{R}$ so that $\dot{R} = f(\tilde{t}) \cdot g(Pe)$ and so that the Peclet dependency takes on the form of $g(Pe) = a \cdot Pe^{-1} + b$, satisfying results were obtained for $a, b = 3.403, 3.984$ ($R^2 = 0.994$). The scaled curves, obtained by dividing the computed curves by the estimated $g(Pe)$, are depicted in the figure inset. The relation Eq. (14) also suggests the reason as to why, at higher Peclet numbers, the Peclet dependency diminishes and the global reaction rate approaches an approximately constant (independent of Peclet) non-zero value. Similarity solutions $\tilde{c} = f(\tilde{x}, \tilde{t}, Pe)$ for Eq. (14) are possible if the normalized velocity $\tilde{v}$ remains identical in the different Peclet number scenarios. In the case of a minor transport reaction interaction that leads to minor conductivity changes, the deviations in the velocity field from the initial value can be assumed to be minor as well; thus, $\tilde{v} \approx 1$. For high Peclet values, the similarity solution to Eq. (14) is approximately Peclet independent. As follows from Fig. 2a, the ratio between the global reaction rate for $Pe = 12$ and $Pe = 0.38$ is about 4, which may hint at the fact that the advective contribution to reaction rate is small compared to the diffusive contribution for the lower range of Peclet numbers in the current study.

Figure 2b presents the normalized global reaction rate $\dot{R}_{\mathrm{norm}}$ that was obtained by normalizing the global reaction rate $\dot{R}$ by the distance from the inlet sampled by the flow $L_x \tilde{t}$ (excluding the area in the vicinity of the inlet). This was done in order to obtain an indication of the reaction rate per unit area where the reaction takes place. After the transitional effects fade, $\dot{R}_{\mathrm{norm}}$ exhibits an approximately constant behavior, as expected. Recall that, on a local scale, in our model, the reaction is instantaneous (Edery et al., 2021). Here, as well, the scalability with the reciprocal of the Peclet num-

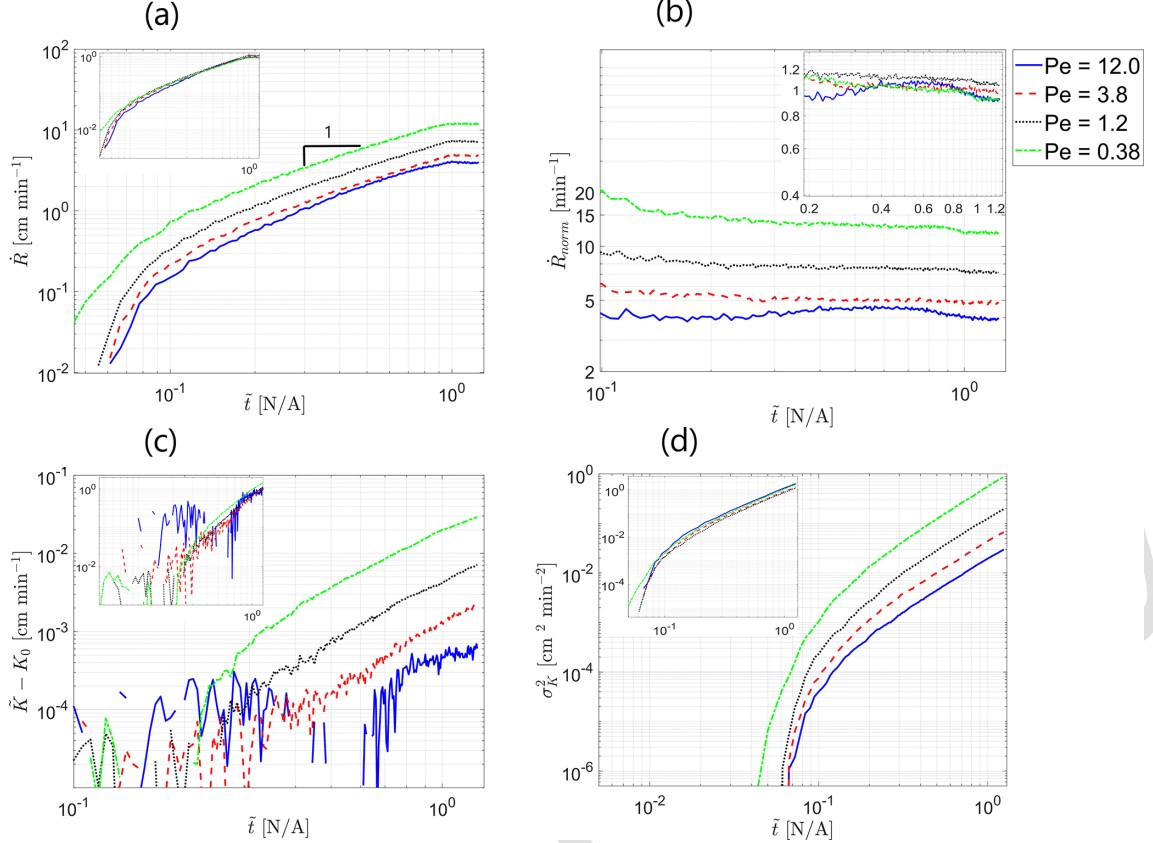

**Figure 2.** Influence of Peclet number on the evolution of the reactive process in the field over time: **(a)** global reaction rate $\dot{R}$, **(b)** normalized global reaction rate $\dot{R}_{\mathrm{norm}}$, **(c)** deviation of the mean hydraulic conductivity from the initial value $\tilde{K} - K_0$, and **(d)** hydraulic conductivity variance $\sigma_k^2$ as a function of dimensionless time $\tilde{t}$. Insets show Peclet-scaled curves, obtained by assuming a power-law Peclet dependency (a dependency on the reciprocal of Peclet was obtained for all displayed parameters as an exponent value of $-1$ resulted in a satisfying fit for all curves).

ber exists, taking the same form as in the case of the non-normalized global reaction rate $\dot{R}$. The scaled curves are depicted in the figure inset.

Figure 2c presents the deviation of the mean value of hydraulic conductivity over all computational cells from the initial conductivity $\tilde{K} - K_0$ as a function of dimensionless time $\tilde{t}$. Here, the area of 10 computational cells in the immediate vicinity of the inlet was excluded from the calculation as well. Clearly, the net reaction is tilted towards dissolution as $\tilde{K} - K_0$ grows monotonously with $\tilde{t}$, which is reasonable due to the influx of low-pH fluid at the inlet of the field (see also Edery et al., 2021); the relative increase in hydraulic conductivity is small compared to $K_0$. This result is in agreement with the findings of Edery et al. (2021). The power-law shape holds well, beginning with approximately $\tilde{t} = 0.3$. The onset of the power-law region coincides with the time when a large-enough particle ensemble has reacted within the field to create a statistically consistent picture. The curves for larger Peclet numbers exhibit fluctuations for a considerable portion of the time; however, they comply with the power-law shape eventually as well. $\tilde{K} - K_0$

also increases with the decrease in the Peclet number, again due to the reaction-enhancing role of the diffusive transport mechanism. Here, as well, the scalability with the reciprocal of the Peclet number exists. Assuming, again, the separation of variables as before so that the Peclet dependency takes on the form $g(Pe) = a \cdot Pe^{-1}$, good results were obtained for $a = 0.007624$ ($R^2 = 0.985$). This can be justified by, again, relating the parameters that govern the evolution of the reactive process in the field to the reaction rate, for which the Peclet dependence was shown to have the same shape. All Peclet-scalability-related conclusions drawn for Fig. 2a are applicable here as well. The scaled curves are depicted in the figure inset.

Similar trends are observed in Fig. 2d, which presents the variance of hydraulic conductivity $\sigma_k^2$ as a function of dimensionless time $\tilde{t}$. Here, the area of 10 computational cells in the immediate vicinity of the inlet was excluded from the calculation as well. This parameter, being the measure of the field's heterogeneity, is indicated as the primary cause for transport self-organization in the field (Zehe et al., 2021). $\sigma_k^2$ increases monotonously with dimensionless time $\tilde{t}$ as

the particles sample more field regions and react at an increasing rate, with this result again being in agreement with the findings of Edery et al. (2021). Heterogeneity also increases with the decrease in the Peclet number, again due to the reaction-enhancing role of the diffusive transport mechanism. We thus state that reactive-transport scenarios with lower values of Peclet number, corresponding to the dominant diffusive transport mechanism, coincide with an increased reaction rate in the field and, thus, increased field heterogeneity. The power-law shape holds well for all Peclet cases, beginning with approximately $\tilde{t} = 0.2$. Turning again to the findings of Edery et al. (2021), where the coupled reactive process was investigated within the framework of the continuous-time random-walk (CTRW) approach, we emphasize the anomalous (non-Fickian) nature of the ensuing transport that is traced back to the coupling between the reactive and transport processes in the initially homogeneous field. Here, as well, the scalability with the reciprocal of the Peclet number exists. Assuming, again, the separation of variables as before so that the Peclet dependency takes on the form $g(Pe) = a \cdot Pe^{-1}$, good results were obtained for $a = 0.001616$ ($R^2 = 0.987$). All Peclet-scalability-related conclusions drawn for Fig. 2a–c are applicable here as well. The scaled curves are depicted in the figure inset.

The dependence of the parameters, depicted in Fig. 2a–d, on the reciprocal of the Peclet number confirms that this dimensionless number is indeed the driving force behind the evolution of the reactive process in the field. The fact that Peclet-related tendencies, obtained in Sect. 3.1 for the simple one-dimensional ADRE scenario with an adsorption reaction case, represented by the linear Freundlich isotherm, have been verified in the Lagrangian particle tracker numerical simulations presented above, where a considerably more complex reversible reaction of dissolution and precipitation of calcite is considered, suggests a general validity of these tendencies for the case of minor heterogeneity variations that lead to minor fluctuations in concentration. An important note should be made regarding the initial state of the field that undergoes the reactive process: in our case of an initially homogeneous field, diffusion acts as an enhancing factor for the emergence of heterogeneity in the field; in an initially heterogeneous case, such as in Al-Khulaifi et al. (2017), diffusion is capable of decreasing the heterogeneity of the porous medium due to its smoothing property.

The relatively small scale of the phenomenon presented in this study must be pointed out. The initial state of the porous medium is completely homogeneous, and the heterogeneity that develops is relatively minor. The trends presented could have been more pronounced quantitatively had the simulation been allowed to run longer. The simulation was stopped soon after pore volume time was reached due to computational time limitations since running the LPT code with a large number of particles consumes considerable computational resources. Had the model been allowed to run longer, the mean conductivity and the heterogeneity trends are ex-

pected to have further increased, while the dependency of the reactive-transport evolution on the reciprocal of Peclet is expected to have persisted as long as the degree of heterogeneity of the field remained moderate (see Sect. 3.1).

## 4.2 Transport self-organization in the field

To characterize the emergence and development of transport self-organization in our model as the dissolution–precipitation reactive processes in the field advance, we adopt a straightforward use of the Shannon entropy, similarly to Zehe et al. (2021), where it was employed in the context of characterizing self-organization in non-reactive flows in heterogeneous groundwater systems. In order to place all results on equal footing, we perform computational non-reactive tracer tests on the snapshots of the hydraulic-conductivity field at different values of the dimensionless time $\tilde{t}$, as described in Sect. 3.2.

The emergence of transport self-organization in the reacting field is evident from the Fig. 3a–d, which present the decimal logarithm of the relative non-reactive tracer concentration $\tilde{c}$ in the snapshots of the field that undergoes a reactive process at $Pe = 0.38$ for different values of the dimensionless time $\tilde{t}$. $\tilde{c}$ is defined as the ratio of the total number of non-reactive particle visitations in a cell to the number of particle visitations in a cell in the equilibrium state (represented by perfect mixing in the field). Here, the number of particle visitations in the equilibrium state is defined by $N_{\text{tot}}^{\text{NR}}/Ny$, which corresponds to the case of a uniform particle injection in the completely homogeneous field, where the same number of particles visit each cell in the field. This parameter is analogous to the non-reactive solute relative concentration obtained from the tracer tests. With the passage of time, variations in the hydraulic conductivity of the field due to the reactive process create an autocatalytic feedback mechanism that leads to an emergence of finger-like preferential flow paths. These paths interact, competing for the available flow, so that eventually some of the paths carry a significantly larger part of the injected particles than the others, as seen from the increasing concentration gradients in the direction transverse to flow. This can be explained by observing the mechanism responsible for transport self-organization in the field. The particles injected at the inlet of the field are flux weighted, meaning that the number of particles injected into each of the inlet cells is proportional to the Darcy flux in that cell. Diffusion, treated as a stochastic agent in our model, leads to the appearance of hydraulic-conductivity fluctuations in the initially homogeneous field in the direction transverse to flow. Thus, the more conductive paths receive more Darcy flux, which in turn attracts more reactive $A$ particles to them, making these paths even more conductive. This autocatalytic process leads to constant intensification of this phenomenon, as shown in Fig. 3a–d. The observed flow paths are linear in shape, with negligible tortuosity, due to the hydraulic-conductivity deviations from the initial value being minor.

Another reason for this is that the correlation length of the produced heterogeneous fields is small as well, leading to an absence of large-scale structures that could alter significantly the direction of the preferential flow paths. This picture corresponds to the simplified hydraulic model presented in Sect. 3.3, where the field was considered as a series of approximately linear conductive channels connected in parallel. The similarities of the simulated scenario to that of reactive infiltration, as reported in Szymczak and Ladd (2006), where a pore-scale numerical model was used to investigate channel growth and interaction due to dissolution in fractures, have to be pointed out. The model mentioned above employed the Lattice–Boltzmann method for flow field calculation, while the transport of dissolved species was modeled by a random-walk algorithm that efficiently incorporates the chemical kinetics at the solid surfaces. Szymczak and Ladd (2006) reported the emergence of a solid–fluid interface instability, with undulation areas formed first at the solid–fluid interface of the porous media due to dissolution, later transformed into well-defined, finger-like channels or wormholes that rapidly advance into the medium; as dissolution proceeds, these fingers interact, competing for the available flow, and eventually the growth of the shorter ones ceases.

The deviations of the decimal logarithm of the relative non-reactive tracer concentration $\tilde{c}$, shown in Fig. 3a–d, from the equilibrium value of 0 represent either the paths where fewer particles pass relative to the equilibrium state (negative values of $\tilde{c}$) or the paths where more particles pass relative to the equilibrium state (positive values of $\tilde{c}$). Considering the temporal evolution of $\tilde{c}$, one clearly sees an intensification of these deviations from the equilibrium value of 0 with time. In Fig. 3d ($\tilde{t} = 1.0$), there are a few paths that correspond to the decimal logarithm of $\tilde{c} = -0.4$ and below (61 % fewer particles that visited the path than in the case of a completely homogeneous particle distribution in the field), while some of the others correspond to the decimal logarithm of $\tilde{c} = 0.2$ and above (58 % more particles that visited the path than in the case of a completely homogeneous particle distribution in the field).

The observations from Fig. 3 are confirmed by the plot of the normalized Shannon entropy of the transport in the field ($S_{\mathrm{norm}} = (S - S_{\mathrm{max}})/S_{\mathrm{max}}$ vs. normalized distance from inlet $\tilde{x} = x/L$), calculated from the non-reactive solute concentration data in the snapshots of the reactive field at different values of dimensionless time $\tilde{t}$ for $Pe = 0.38$, presented in Fig. 4a. Here, $S$ is the Shannon entropy of the transport in the field as a function of the distance from inlet $X$, calculated from Eq. (16), and $S_{\mathrm{max}}$ is the maximum possible transport entropy value in the field, calculated as discussed in Sect. 3.2, so that $S_{\mathrm{norm}}$ obtains values from 0 (maximum entropy, no self-organization) to $-1$ (minimum entropy, maximum self-organization, corresponding to a single preferential flow path channeling all injected solute particles). As the reactive process in the field advances, the resulting level of transport self-organization in the field increases as it reaches further downstream alongside the reaction front, which is signified by the decrease in $S_{\mathrm{norm}}$. Figure 4b presents the mean value of the normalized Shannon entropy of the transport in the field, averaged over the field, $\tilde{S}_{\mathrm{norm}}$, vs. dimensionless time $\tilde{t}$ for different values of Peclet number. Again, the plot clearly shows an increase in the transport self-organization in the field with the advance of the reactive process as the mean normalized entropy $\tilde{S}_{\mathrm{norm}}$ decreases over time. A clear picture emerges where the level of self-organization directly correlates with the Peclet number of the reactive flow as the normalized entropy $\tilde{S}_{\mathrm{norm}}$ decreases with a decrease in Peclet number. This clearly signifies that the diffusive transport mechanism is dominant in initiating transport self-organization in an initially homogeneous field via the coupled reactive process as it is manifested most significantly for low Peclet numbers that correspond to diffusion-dominated flow. For lower values of the Peclet number, the difference between the neighboring curves is clearly distinguished, while for larger Peclet numbers, where advection becomes dominant, this difference diminishes, further supporting our discussion on limiting cases for the model under investigation that appears in Sect. 3.1.

In this context, it is interesting to consider the findings of Zehe et al. (2021), which showed a clear correlation between the field's heterogeneity measure (statistical variance of the field's conductivity) and the self-organization of the field's transport, as represented by the Shannon entropy of the solute concentration matrix, in non-reactive transport in heterogeneous fields. Their results showed a clear decrease in Shannon entropy following an increase in the field's heterogeneity. These results are given further confirmation in our study as, with the passage of time, the development of the reactive process in an initially homogeneous field results in an increase in its heterogeneity (Fig. 2b), accompanied by a decrease in the Shannon entropy of the transport in the field that corresponds to an increase in transport self-organization (Fig. 3). To conclude our findings thus far, we state that the reactive transport at lower Peclet numbers, corresponding to diffusion-dominated flow, results in a higher field heterogeneity and, thus, stronger transport self-organization in the field. An important note should be made regarding the initial state of the field that undergoes the reactive process: in our case of an initially homogeneous field, diffusion acts as a self-organization enhancer; in an initially heterogeneous case, diffusion is capable of decreasing the transport self-organization due to its smoothing property.

The Peclet number scalability is also exhibited in the Shannon entropy of transport self-organization. Assuming again the separation of variables in $\tilde{S}_{\mathrm{norm}}$ so that $\tilde{S}_{\mathrm{norm}} = f(\tilde{t}) \cdot g(Pe)$ and so that the Peclet dependency takes on the form $g(Pe) = a \cdot 2^{-\cdot Pe} + b$, satisfying results were obtained for $a, b = -0.02665, -0.000273$ ($R^2 = 0.9942$). Thus, the mean normalized Shannon entropy of transport self-organization depends on the reciprocal of 2 to the power of the Peclet number. The free coefficient $c$ is expected to be

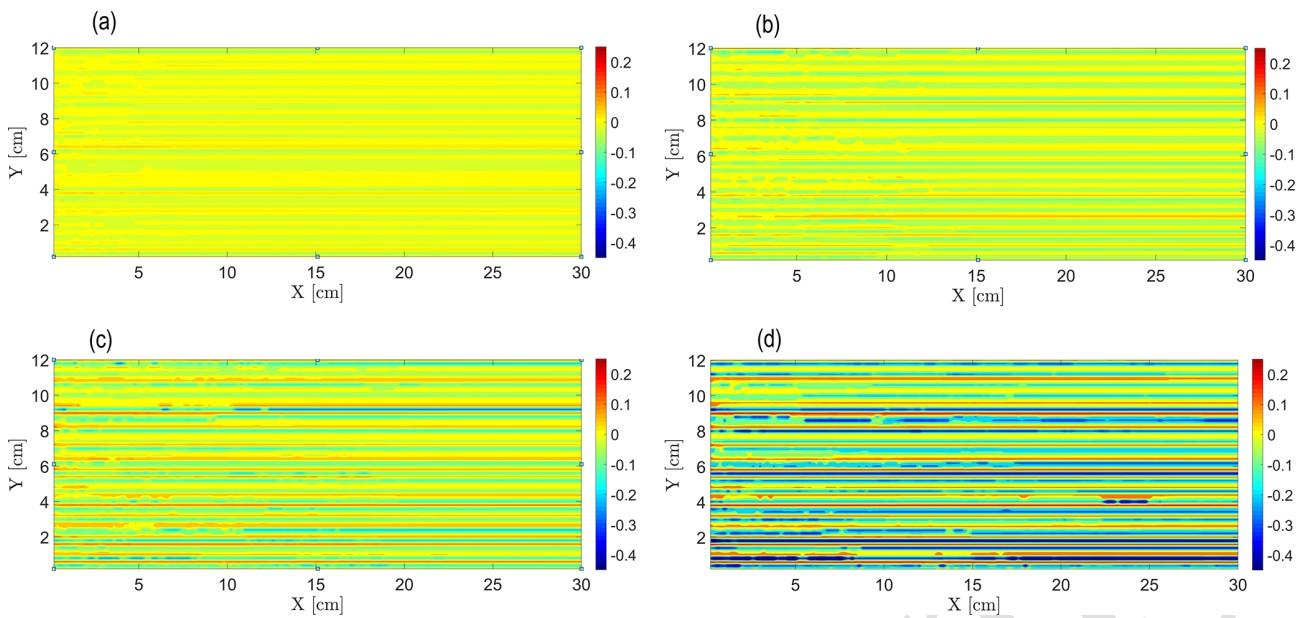

**Figure 3.** Evolution of the transport self-organization in the field for $Pe = 0.38$, as represented by the decimal logarithm of the relative non-reactive tracer concentration $\tilde{c}$ based on data obtained from the non-reactive tracer tests performed on the snapshots of the field over time: **(a)** $\tilde{t} = 0.25$, **(b)** $\tilde{t} = 0.5$, **(c)** $\tilde{t} = 0.75$, and **(d)** $\tilde{t} = 1.0$.

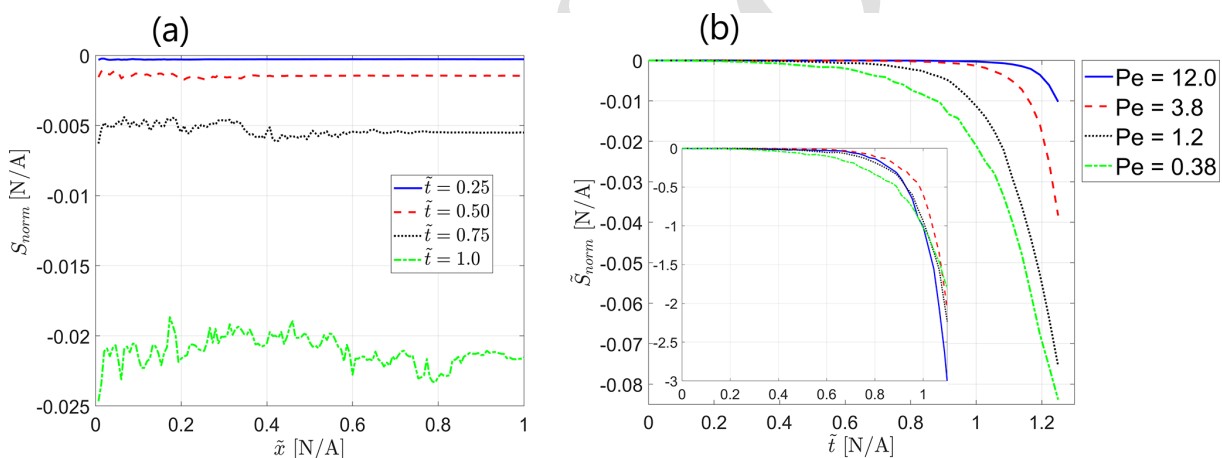

**Figure 4.** Self-organization of the transport in the field via normalized Shannon entropy $S_{norm} = (S - S_{max})/S_{max}$: **(a)** $S_{norm}$ vs. normalized distance from inlet $\tilde{x} = x/L$ at different dimensionless times $\tilde{t}$ for $Pe = 0.38$, **(b)** mean value of the normalized entropy over the field $\tilde{S}_{norm}$ vs. dimensionless time $\tilde{t}$ for different values of $Pe$. Inset shows Peclet-scaled curves, obtained by assuming a power-law Peclet dependency.

zero since, in the high-Peclet limit, we expect to obtain no transport self-organization in the field; deviation from this value is attributed to statistical errors that can be decreased by averaging results from several realizations of the reactive
5 process. The scaled curves are depicted in the figure inset. This dependence of the mean normalized Shannon entropy of transport self-organization on the Peclet number confirms that this dimensionless number is indeed the driving force behind the transport self-organization in an initially homo-
10 geneous porous medium. The presented Peclet dependency trends are reminiscent of the parameters that characterize

the evolution of the reactive process in the field, analyzed in Sect. 4.1. They can be justified by directly relating the transport self-organization in the field to heterogeneity, which in turn is related to the global reaction rate in the field, for which 15 the Peclet dependence was shown in an analytical result, derived in Sect. 3.1 (see also discussion in Sect. 4.1).

Here, again, the relatively small scale of the presented phenomenon must be pointed out, although the apparent tendencies are clear. Had the model been allowed to run longer, a 20 further increase in heterogeneity would lead to a more pronounced decrease in the Shannon entropy of the transport.

The dependency on the reciprocal of the Peclet number is expected to persist as long as the degree of heterogeneity of the field remains moderate (see Sect. 3.1). The incompleteness of the Shannon entropy as a measure of transport self-organization in the field should be pointed out since it does not take into account spatial correlations that may be present in the transport configuration in the field. However, it might be accurate enough for the specific case of an initially homogeneous field, as presented in the paper, since, in this case, the self-organization mimics the form of straight parallel channels from inlet to outlet. Thus, spatial correlation will not bear any additional information in this case as no larger-scale structures are expected to emerge in the field.

## 4.3 Entropy export into the breakthrough curve

As a next stage in the analysis of transport self-organization in the reactive field, we turn to the particle breakthrough curve in an attempt to analyze the influence of self-organization of the field's transport on the particle arrival times. Figure 5a shows the temporal breakthrough curves calculated from the non-reactive particle-tracking tests performed on the snapshots of the reacting field for $Pe = 0.38$, as described in Sect. 3.2. The arrival times $T_{BT}$ are normalized by the pore volume time $T_{pv}$. The width of support of the breakthrough curves clearly increases with time, indicating an increasing scatter in non-reactive particle arrival times as the field's heterogeneity grows larger. In particular, a significant increase in the tailing arrival times is observed.

The Shannon entropy of the breakthrough curves was calculated by dividing the arrival time span into 1000 bins and employing Eq. (16) with $p_i$ as the relative occurrence of arrival times. To place all results on equal footing, the arrival time span was taken to be identical for all Peclet values. Figure 5b presents the normalized breakthrough curve entropy, calculated as $S_{norm}^{BTC} = (S^{BTC} - S_{max})/S_{max}$, vs. dimensionless time $\tilde{t}$. Here, $S^{BTC}$ is the breakthrough curve entropy, calculated from Eq. (16), and $S_{max}$ is the maximum possible entropy value, obtained in the case of a perfectly uniform distribution of arrival times (similarly to the discussion of $S_{max}$ estimation for the field transport entropy held in Sect. 3.2, in the case of division of the arrival times span into 1000 bins, this value will be $\log_2 1000$). Again, $S_{norm}^{BTC}$ obtains values from 0 (no self-organization) to $-1$ (maximum self-organization, corresponding to the case when all particles arrive at the outlet at the same time).

A picture emerges where the Shannon entropy of the arrival times increases with the passage of time, reflecting a larger uncertainty and a declining order in the temporal distribution of travel times. This corresponds to an increase in both the field's heterogeneity and the self-organization of transport as the reactive process in the field evolves. We also observe that the level of self-organization in arrival times directly correlates with the Peclet number as the normalized breakthrough entropy $S_{norm}^{BTC}$ decreases with an increase in

Peclet number, an opposite tendency compared to the entropy of the field transport (see Sect. 4.2).

The Peclet number scalability is also exhibited in the Shannon entropy of the breakthrough curve, from approximately $\tilde{t} = 0.5$. Assuming again the separation of variables in $S_{norm}^{BTC}$ so that $S_{norm}^{BTC} = f(\tilde{t}) \cdot g(Pe)$ and so that the Peclet dependency takes on the power-law form $g(Pe) = a \cdot Pe^b + c$, satisfying results were obtained for $a, b, c = 0.5511, -0.1686, -1.0$ ($R^2 = 0.9762$). The free coefficient $c$ obtains the value of $-1$ because, in the high-Peclet limit, we expect to obtain maximal self-organization in the breakthrough times. The trends presented in Fig. 5b are further supported by Fig. 5c, which shows the normalized field transport entropy $\tilde{S}_{norm}$ as a function of the normalized breakthrough curve entropy $S_{norm}^{BTC}$ for different values of Peclet number: $\tilde{S}_{norm}$ decreases monotonously with an increase in $S_{norm}^{BTC}$ for all values of Peclet number. These results can be explained straight ahead in terms of the degree of the field's heterogeneity: for lower Peclet values, the reactive transport in the field leads to a higher degree of heterogeneity, which is responsible for a greater scattering in arrival times due to the variety of paths particles may take on their way to outlet; this is reflected in the overall smoothing out of the breakthrough curve and, as a result, in the breakthrough curve entropy that grows with the degree of scattering in arrival times. While there are other factors that affect the total entropy budget of the problem, such as production of the macroscopic-flow entropy due to hydraulic power dissipation through heat, these two properties are correlated in a self-consistent way. Thus, a thermodynamic analogy can be made: since, according to the second law of thermodynamics, the overall entropy of the system and its surroundings cannot decrease, the decreasing entropy of the system, represented, among other contributions, by the entropy of the transport in the field, needs to be exported outside. This leads to an increase in the entropy of the surroundings, which is reflected, among other factors, in an increase in the temporal breakthrough curve entropy (see Sect. S1). This analogy should be regarded on a qualitative level only.

## 4.4 Hydraulic power – entropy relation

The evolution of the reactive process in an initially homogeneous field is accompanied by the emergence of heterogeneity and, consequently, transport self-organization in the field. This demands that increased energy is supplied to the field to maintain an increasingly ordered state that emerges there. This energy is supplied in the form of hydraulic power, required for the flow to overcome the hydraulic resistance of the porous medium. In our model, the hydraulic power is supplied due to the inlet–outlet hydraulic-head-drop boundary condition applied to the porous medium, which works similarly to a pump.

Figure 6a presents the normalized hydraulic power in the field $(P - P_0)/P_0$ as a function of dimensionless time

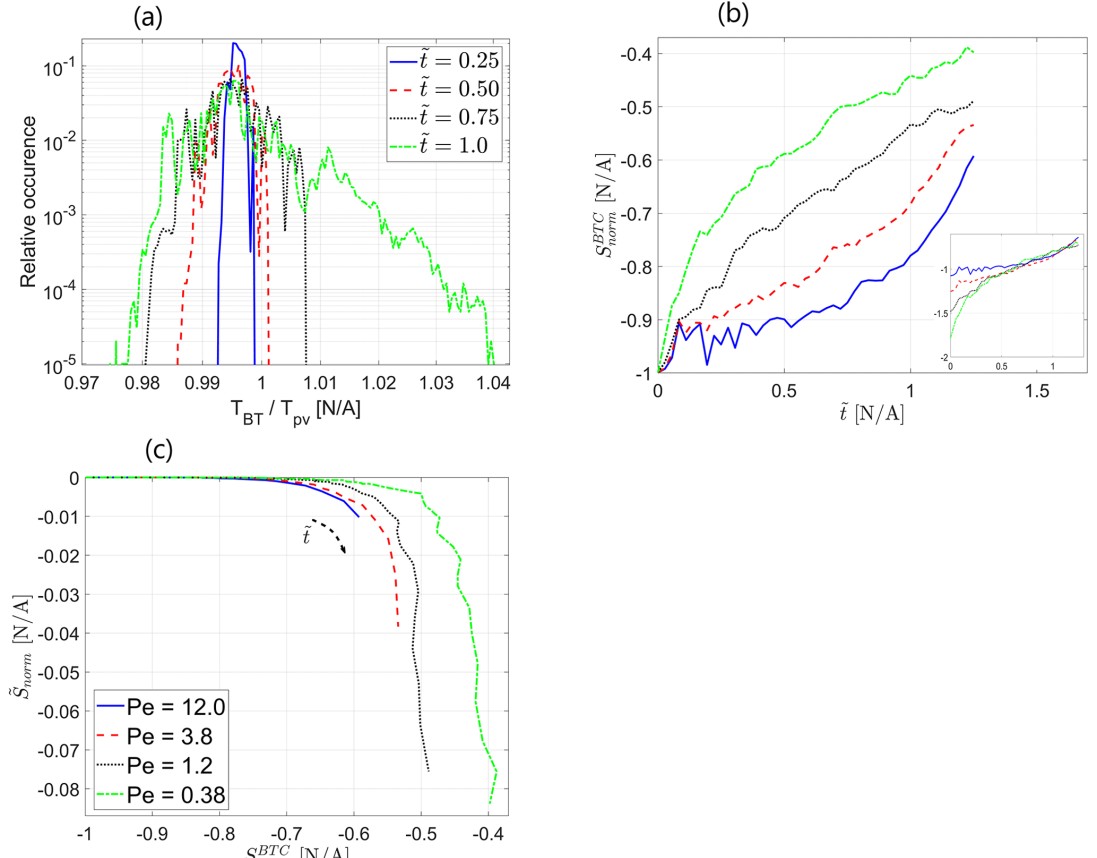

**Figure 5.** Breakthrough curve self-organization: **(a)** histogram of the non-reactive particle arrival times $T_{\mathrm{BT}}$, normalized by the pore volume time $T_{\mathrm{pv}}$, in snapshots of the reacting field at different dimensionless times $\tilde{t}$ for $Pe = 0.38$; **(b)** normalized breakthrough curve entropy $S_{norm}^{\mathrm{BTC}}$ vs. dimensionless time $\tilde{t}$ for different values of Peclet number (the inset shows Peclet-scaled curves); and **(c)** normalized field transport entropy $\tilde{S}_{norm}$ vs. normalized breakthrough curve entropy $S_{norm}^{\mathrm{BTC}}$ for different values of Peclet number. Time direction is specified by an arrow.

$\tilde{t}$ for different Peclet numbers, calculated using different approaches (see Sect. 3.3). Here, the unmarked lines represent cell-wise power calculation ($P_{\mathrm{total}}$, according to Eq. 17), while the lines marked with circular markers represent power, calculated based on the parallel-channels model ($P_{\mathrm{total}}^{\mathrm{EQ}}$, according to Eq. 20). Due to the linearity of the observed preferential flow paths, as presented in Fig. 3, the channels in the simplified model used to calculate $P_{\mathrm{total}}^{\mathrm{EQ}}$ were assumed to be of a linear shape; thus, the indices $i, j$ in Eq. (19) correspond to the rows and columns of the computational field. In both approaches, the power is calculated over the whole field. From this comparative plot, it is evident that, for all Peclet numbers, the power increases with $\tilde{t}$, along with the increase in heterogeneity of the field, and that is in accordance with our expectations as detailed in Sect. 3.3. We also observe that the parallel-channel assumption is valid as a rule in the initial part of the reactive process; however, as the normalized time $\tilde{t}$ grows, the deviations from this assumption increase. This can be explained by the fact that

$P_{\mathrm{total}}^{\mathrm{EQ}}$, calculated under this assumption, takes into account only the power produced by the horizontal Darcy flux component since the parallel-channel model does not allow interaction between adjacent channels, while $P_{\mathrm{total}}$, calculated in a cell-wise fashion, accounts also for the transverse (vertical) flux component. With the passage of time, due to an increase in the heterogeneity of an initially homogeneous field, flux in the transverse direction appears. With time, its contribution grows as it appears more significant. We observe that, until about $\tilde{t} = 0.4$, the parallel channel assumption appears to be adequate for all Peclet values. Beginning from $\tilde{t} = 0.4$, deviations from this assumption are visible. The magnitude of these deviations is in direct correlation with the Peclet number and increases with the decrease in Peclet number. This can be explained again by virtue of heterogeneity, which increases with the decrease in Peclet number, as well as with $\tilde{t}$.

The confirmation of the proposition put forth in Sect. 3.3, which relates the increase in the total hydraulic power dissi-

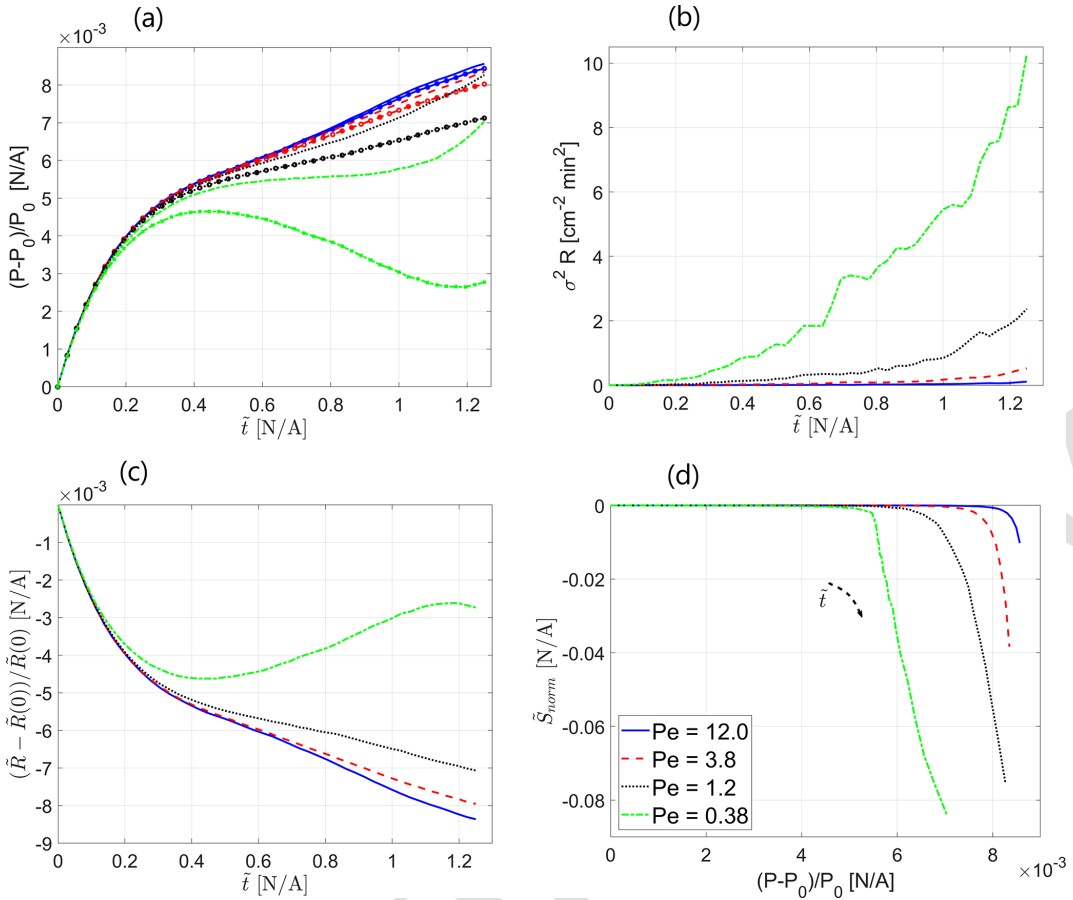

**Figure 6.** Hydraulic power in the field and its relation to heterogeneity and transport entropy: **(a)** normalized hydraulic power vs. dimensionless time $\tilde{t}$ (unmarked lines represent cell-wise power calculation $P_{\text{total}}$, while the lines marked with circular markers represent power, calculated based on the parallel channels model $P_{\text{total}}^{\text{EQ}}$); **(b)** variance of the channel hydraulic resistivities $\sigma^2 R$ vs. dimensionless time $\tilde{t}$; **(c)** mean value of the channel hydraulic resistivities $\tilde{R}$, normalized by the value at $\tilde{t} = 0$ vs. dimensionless time $\tilde{t}$; and **(d)** normalized mean field transport entropy $\tilde{S}_{\text{norm}}$ vs. normalized hydraulic power $(P - P_0)/P_0$ for different values of Peclet number. Time direction is specified by an arrow.

pated in the field to an increase in heterogeneity of the field by virtue of the simplified parallel-channels model, is given in Fig. 6b, which presents the variance of the hydraulic resistivities of the channels $\sigma^2 R$ vs. dimensionless time $\tilde{t}$. We observe that $\sigma^2 R$ increases monotonously with $\tilde{t}$ as the overall field heterogeneity grows larger, signifying the underlying relation between heterogeneity and the dissipated power in the field. It is also interesting to notice that, although the increase in the heterogeneity of the field is largest for the lower values of the Peclet number, the normalized hydraulic power, presented in Fig. 6a, grows faster for higher Peclet values. The explanation for this lies in the fact that, along with increasing the heterogeneity of the field, the coupled reactive dissolution–precipitation process also works to increase the mean value of the hydraulic conductivity in the field (see Sect. 4.1). This also contributes to an increase in the total hydraulic power in the field, as can be seen from Eq. (21). The confirmation for this can be seen in Fig. 6c, which presents

the mean value of the channel hydraulic resistivities $\tilde{R}$, normalized by the value at $\tilde{t} = 0$, vs. dimensionless time $\tilde{t}$. We observe that $\tilde{R}$ decreases with time, meaning that the channels' mean conductivities grow larger; this is in accordance with Fig. 2c. However, we also observe that faster decrease rates are observed for the higher Peclet values, which contradicts the trends presented in Fig. 2c. This apparent contradiction is due to the fact that, in the hydraulic power calculation, the entire field was taken into account as opposed to parameters presented in Fig. 2, where an area of 10 computational cell columns in the immediate vicinity of the outlet was left out to exclude the effect of the dissolution area there. Thus, when including the dissolution area at the inlet in the calculation, we observe the Peclet dependency trends overturn; at the inlet, more dissolution occurs for the higher Peclet values due to a clear bias towards dissolution since no $B$ particles are present there. Thus, the behavior of the normalized hydraulic power in the field (Fig. 6a) is controlled by the com-

bined effect of two parameters: the variance of the channel hydraulic resistivities $\sigma^2 R$ (Fig. 6b) and the mean value of the channel hydraulic resistivities $\tilde{R}$ (Fig. 6c).

Finally, Fig. 6d presents the normalized mean field transport entropy $\tilde{S}_{\mathrm{norm}}$ as a function of the net hydraulic power $(P - P_0)/P_0$ for different values of the Peclet number. This plot shows the expected relation between the entropy of the transport and the normalized hydraulic power in the field: for each Peclet number, the entropy of the transport in the field decreases with an increase in the normalized hydraulic power, reflecting an increase in transport self-organization due to an increase in power, supplied to the field in order to maintain the prescribed hydraulic-pressure-drop boundary condition over the field.

## 5  Conclusions

Our computational study tackles the quantitative characterization of the transport self-organization that emerges in an initially homogeneous calcite porous medium due to the dynamic interaction between the reactive precipitation–dissolution processes and the solute transport. Our work leads to the following key conclusions:

  – As the reactive particles advance and react in an initially homogeneous field, heterogeneity is introduced into the field in the form of local dissolution–precipitation areas, evolving further downstream and intensifying with time. A dissolution area is located in the immediate vicinity of the inlet, where the chemical equilibrium is tilted towards dissolution; this correlates with experimental and simulation observations. The global reaction rate, normalized by the distance from the inlet sampled by the flow, is shown to be approximately constant in time. The influence of the Peclet number is exhibited by an increase in the global reaction rate with the decrease in Peclet, thus confirming that the diffusive transport mechanism, responsible for the mixing in the field, enhances the reactive process. This can be explained by the fact that diffusion, being stochastic in nature, allows particles to sample regions away from the path suggested by the advection mechanism, thus allowing a better chance for reaction. The net reaction is tilted towards dissolution as the mean conductivity value in the field grows monotonously with time, as well as with the Peclet number, which is reasonable due to the influx of low-pH fluid at the inlet of the field. Similar trends are observed in the evolution of the variance of hydraulic conductivity in the field. Scalability of the parameters that characterize the evolution of the reactive process in the field with the reciprocal of the Peclet number was derived using a simple one-dimensional ADRE model with a linear adsorption reaction term and then confirmed through numerical simulations, with the global reaction rate, the mean value, and the variance of the hydraulic-conductivity distribution in the field all exhibiting dependency on the reciprocal of the Peclet number. We thus state that reactive-transport scenarios for lower Peclet values, corresponding to the dominant diffusive transport mechanism, coincide with an increased global reaction rate in the field and, thus, an increase in the field's heterogeneity. The dependence of the parameters that characterize the reactive process in the field on the reciprocal of the Peclet number confirms that this dimensionless number is indeed the driving force behind the evolution of the reactive process in the porous medium.

  – As the reactive processes in the field advance, the evolving variations in the hydraulic conductivity of the field create an autocatalytic feedback mechanism that leads to an emergence of finger-like preferential flow paths of a linear shape. These paths interact, competing for the available flow, so that eventually some of the paths carry a significantly larger part of the injected particles than the others, as seen from the increasing concentration gradients in the direction transverse to flow. This observation is confirmed by the mean normalized Shannon entropy in the field that decreases with time, signifying an increase in the level of transport self-organization in the field. The influence of the Peclet number on the evolution of transport self-organization is exhibited by a decrease in the mean normalized Shannon entropy of the transport with a decrease in Peclet number, signifying an increase in self-organization. This clearly indicates that diffusion is the dominant mechanism in creating self-organization in the initially homogeneous field. Here the link between the emergence of heterogeneity and transport self-organization in an initially homogeneous field is being drawn as the increase in heterogeneity results in the increase in the level of self-organization of the transport in the field. Peclet number scalability was shown for the mean normalized Shannon entropy curves as well. To conclude, we state that reactive transport at lower Peclet numbers, corresponding to diffusion-dominated flow, results in a higher field heterogeneity and, thus, stronger transport self-organization in the field. An important note should be made regarding the initial state of the field that undergoes the reactive process: in the case of an initially homogeneous field, diffusion acts as self-organization enhancer; in an initially heterogeneous case, diffusion is capable of decreasing the transport self-organization due to its smoothing property.

  – To switch the observing perspective, the particle temporal breakthrough curve was analyzed in an attempt to understand the influence of self-organization of the field's transport on the particle arrival times. The width of support of the breakthrough curve clearly increases with time as the reactive process in the field evolves,

indicating an increasing scatter in arrival times of non-reactive particles; this result is in line with an increase in the field's heterogeneity. A picture emerges where the Shannon entropy of the arrival times increases with the passage of time, reflecting a larger uncertainty and a declining order in the temporal distribution of travel times. We also observe that the level of self-organization in arrival times directly correlates with the Peclet number as the normalized breakthrough curve entropy decreases with an increase in Peclet number, an opposite tendency compared to the entropy of the field transport. While there are other factors that affect the total entropy budget of the problem, such as production of the macroscopic-flow entropy due to hydraulic power dissipation through heat, these two properties are correlated in a self-consistent way. Thus, a thermodynamic analogy can be made: since, according to the second law of thermodynamics, the overall entropy of the system and its surroundings cannot decrease, the decreasing entropy of the system, represented, among other contributions, by the entropy of the transport in the field, needs to be exported outside. This leads to an increase in the entropy of the surroundings, which is reflected, among other factors, by an increase in the temporal breakthrough curve entropy. This analogy should be regarded on a qualitative level only.

– The evolution of the reactive process in an initially homogeneous field, accompanied by the emergence of heterogeneity and, consequently, transport self-organization in the field, demands that increased energy is supplied to the field to maintain the increasingly ordered state that emerges therein. This energy is supplied in the form of hydraulic power, required for the flow to overcome the hydraulic resistance of the porous medium. We observe that, for all Peclet numbers, the total dissipated hydraulic power increases with time, along with the increase in heterogeneity and mean hydraulic conductivity of the field, confirming the trends suggested by a simple parallel-channels model. The entropy of the transport in the field decreases with an increase in the hydraulic power, reflecting an increase in transport self-organization. This increasing power is supplied to the field in order to maintain the hydraulic-head-drop boundary condition due to an increase in the field's heterogeneity. Following our thermodynamic framework, we thus argue that the power, required to maintain the driving head pressure drop boundary condition between inlet and outlet, increases with the increasing variance of the hydraulic conductivity in the field due to the evolution of the reactive process therein (although the contribution of an increase in the mean conductivity value also should not be forgotten in this context). This increase in the supplied power results in

an increase in the level of transport self-organization in the field.

– To conclude, the scenario presented in the paper corresponds to that of an open thermodynamic system that interacts with its surroundings by exchanging matter and energy. Due to the influx of power from outside, this system is kept in a non-equilibrium state that corresponds to a certain degree of internal self-organization and, thus, to a decreased entropy state. This decrease in the entropy of a system corresponds to an increase in the entropy of the surroundings by means of production of entropy in the system through various processes, which is later exported outside. In our system, we investigate the interplay between three thermodynamic parameters: the entropy of transport self-organization within the field; the entropy of the breakthrough curve; and the hydraulic power, dissipated by the flow, which may be viewed as an influx of power required to maintain the current level of the system's self-organization. While there are additional processes that may influence the overall entropy budget, such as production of the macroscopic-flow entropy due to hydraulic power dissipation through heat, we find that these three properties are correlated in a self-consistent way. The contributions of the above-mentioned additional factors are beyond the scope of the current study. The increase in the hydraulic power, dissipated in the field, and the emergence of transport self-organization are both the result of an increase in the heterogeneity of the field (although the contribution of an increase in the value of the mean hydraulic conductivity to the hydraulic power also should not be overlooked). The latter, in turn, can be viewed as a consequence of the energy invested in the field by the dissolution–precipitation reactive process. We emphasize that the presented study does not intend to construct a complete thermodynamic formalism for the problem under investigation. The thermodynamic framework presented in the current study is aimed at providing qualitative dependencies and/or trends between the above-mentioned parameters of interest. It is our intent to arrive at a more complete thermodynamic formalism for the reactive flow in porous media in the course of the research work.

– Employing a thermodynamic framework to investigate the dynamic reaction–transport interaction in porous media may prove to be beneficial whenever the need exists to establish relations between the intensification of the preferential flow path phenomenon, represented by the decline in Shannon entropy of the transport, with the amount of reaction that occurred in the porous medium and the change in its heterogeneity. This can be of considerable significance to the implications of reactive-transport interaction in various geophysical applications and can assist, for example, in the estimation of efficient

ways to remediate soil contamination, the determination of optimal conditions for $CO_2$ sequestration, and the estimation of oil extraction rates the boreholes.

*Code and data availability.* Codes and data are available on a dedicated GitHub repository upon request to Evgeny Shavelzon (eshavelzon@campus.technion.ac.il).

*Supplement.* The supplement related to this article is available online at: https://doi.org/10.5194/hess-28-1-2024-supplement.

*Author contributions.* ES developed the research methodology, wrote major parts of the simulation code, performed numerical simulations, analyzed data, and wrote the paper. YE supervised the implementation of the research methodology, supplied the initial version of the simulation code, and oversaw the writing of the paper.

*Competing interests.* The contact author has declared that neither of the authors has any competing interests.

ther geographical representation in this paper. While Copernicus Publications makes every effort to include appropriate place names, the final responsibility lies with the authors.

*Financial support.* This research has been supported by the Israel Science Foundation (grant no. 801/20).

*Review statement.* This paper was edited by Harrie-Jan Hendricks Franssen and reviewed by three anonymous referees.

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
