# Peer review of "Shannon Entropy of Transport Self-Organization Due to Dissolution/Precipitation Reaction at Varying Peclet Number in an Initially Homogeneous Porous Media"

_Hydrology and Earth System Sciences, 2023_

## Author Comment (AC2)

Reviewer: In this paper the authors study the formation of heterogeneities of conductivities in an initially homogeneous porous media as it undergoes a precipitation/dissolution dynamic, during which some part of the media reduces their porosity and hence their permeability while other increase. To characterize the consequences of the heterogeneities of conductivity upon the transport, they choose to consider the entropy of transport quantities, namely (i) concentration of transported non-reactive particles undergoing advection and diffusion, (ii) arrival times of these particles.

Overall, the phenomenon itself of emergence of heterogeneity during this process is of interest. The document is clearly written except for a few rare mistakes (see below), however the figures could be improved and the global articulation of the concepts seems a bit tenuous despite each of them being well explained separately. At the end of the article, I am still left wondering why entropy was chosen to characterize this interesting heterogeneity of conductivity. I propose a solution to the authors, which is to compare their results to reference situations, hydrodynamical situations of thermodynamical situations for which the reader could build his intuition.

Global comments: figures could use some work, validity is good, English is excellent.

I think we can discuss essentially the figures, which follow the development of the argument.

We thank the reviewer for recognizing the quality of our work and for identifying its relevance to the HESS readership. The comments from the reviewer were extremely helpful in clarifying and improving our study, and were pivotal in addressing the identified scaling in our study. These comments also aided in enhancing our figures. We are grateful to the reviewer for the time invested in reviewing our work. The reviewer's comments are marked in blue, while our replies are marked in black. We believe that we have addressed all the comments, which have significantly improved our work.

**1. Figure 1**

1.1 set the aspect ratio of X and Y to 1? daspect([1, 1, 1])

[Figure]

Figure 1: Evolution of the relative hydraulic conductivity $K - K_0$ [$cm\ min^{-1}$] field over time for *Pe = 0.38*: (a) $\tilde{t} = 0.25$, (b) $\tilde{t} = 0.5$, (c) $\tilde{t} = 0.75$, (d) $\tilde{t} = 1.0$.

We thank the reviewer for suggesting ways to improve readability of the manuscript. Aspect ratio has been updated. Please find attached the updated Figure 1. The snapshots of the conductivity field have been taken at normalized times $\tilde{t} = 0.25, 0.5, 0.75, 1.0$ (as opposed to the original manuscript draft where snapshots at $\tilde{t} = 0.5, 0.72, 0.94$ were used). Following the reviewer's suggestion, Figure 1d (dissolution/precipitation areas at $\tilde{t} = 1.0$) has been removed, as it does not carry any essential information. Four snapshots of the conductivity field taken at normalized times $\tilde{t} = 0.25, 0.5, 0.75, 1.0$ have been used instead.

1.2 Plot, on the same plot, the Y averaged conductivity versus X, (++ within its standard deviation?)

[Figure]

Mean (a) and Variance (b) of the Hydraulic conductivity (vertically in the field) vs. Longitudinal X-coordinate

We thank the reviewer for suggesting further verification of the presented results. Please find above the plots of Mean and Variance of vertical rows of the hyd. conductivity field (along Y axis) versus X coordinate for normalized times $\tilde{t} = 0.5, 0.75, 1.0$. The plots show that conductivity fluctuations, approximately symmetrical about the initial value, grow with time. In general, these fluctuations' amplitude is bigger near the inlet, where more reaction has occurred than further downstream of the field. This further confirms the observations from Figure 1.

Although the above plots are important, as they serve for further verification for results presented in Figure 1, it is our opinion that they will not add any essential information to the paper, therefore we suggest that they remain in this document or added to the paper appendix if required.

1.3 It seems the reactive front has not gone through the whole media: could the authors show a map where this is the case, probably for t*>1?

In addition to the updated Figure 1d, which present the reactive field snapshot at $\tilde{t} = 1.0$, please find attached the hyd. conductivity snapshot at $\tilde{t} = 1.1$. The map includes streamlines, obtained with the help of *streamline* MATLAB function (see discussion on streamlines further in the document). Beginning with $\tilde{t} = 1.0$ and further, we observe clearly that the reactive front has arrived at the outlet, as shown by conductivity fluctuations at the outlet that appear as a result of the reactive process in the field.

[Figure]

Relative hydraulic conductivity field $K - K_0 \, [cm \, min^{-1}]$ for Pe = 0.38 at $\tilde{t} = 1.1$ (including flow-based streamlines)

1.4 Concerning the non-dimensionalization of time, maybe the author could plainly state that "qualitatively, t is nondimensionalized so that for t* = 1, the reactive front has permeated through the whole media (cf figure 2 a).

The manuscript draft will be amended accordingly: *The normalized time $\tilde{t}$ is defined in such a way that for $\tilde{t} = 1.0$ the reactive front has permeated through the whole media.*

1.5 Units of K are not specified here. Add them or maybe switch to (K-K0)/K0 to convince us that these changes are indeed minor as stated in the text? Maybe a check of tortuosity would be helpful to convince the readers that the flow remains truly along the x direction?

We sincerely thank the reviewer for noticing this error on our part. Units of the relative hydraulic conductivity are $K - K_0 \, [cm \, min^{-1}]$. This will be specified in the caption for Figure 1. Since the initial hydraulic conductivity of the field is about 11 $[cm \, min^{-1}]$, one can easily see that the changes in the hydraulic conductivity are indeed minor.

A simple check to ensure that the flow remains along the x-direction is by looking at a typical streamlines plot (see the hyd. conductivity map with streamlines for $\tilde{t} = 1.1$ attached again below for your convenience). It is clear from the streamlines direction that the flow remains along the x-direction, while the deviations from homogeneity account only for slight deviations from straight horizontal streamlines (can be visible while zooming in the inlet region, see below).

[Figure]

Relative hydraulic conductivity field $K - K_0$ $[cm\,min^{-1}]$ for Pe = 0.38 at $\tilde{t} = 1.1$ (including flow-based streamlines)

As the deviations between hydraulic conductivities are small, as well as the correlation length of the hyd. conductivity field, the streamlines' deviation from linearity is negligible in tortuosity terms. Zoom in on a single streamline (see plot below) clearly shows that the deviations from the linear path are minor, and the tortuosity here is just about 1.

[Figure]

Relative hydraulic conductivity field $K - K_0$ $[cm\,min^{-1}]$ for Pe = 0.38 at $\tilde{t} = 1.1$ (including flow-based streamlines) – zoom in

We do expect that tortuosity will increase as the reactive process will advance further, yet currently it is out of the scope of this study.

1.6 How come K-K0 is not 0 on the right-hand side of a and b?

We thank the reviewer for noticing this discrepancy. This was an issue of a small numerical deviation from zero due to round-off error, fixed now (see the updated Figure 1 above).

1.7 If you decide to keep (d), get rid of the ticks of the colorbar and mark both colors as dissolving/precipitating areas.

Following the reviewer's suggestion, Figure 1d (dissolution/precipitation areas at $\tilde{t} = 1.0$) has been removed, as it does not carry any essential information. Four snapshots of the conductivity field taken at normalized times $\tilde{t} = 0.25, 0.5, 0.75, 1.0$ have been used instead. The manuscript will be amended accordingly.

**2. Figure 2**

2.1 There are two regimes for the global reactivity and only one for the evolution of the conductivity: how come?

The reviewer is correct in pointing out that there are two plots in Figure 2 that are dedicated to the global reaction rate (2a and 2b), while only one plot is dedicate to the evolution of the mean conductivity value (1c) and one for the variance of the conductivity (1d). The global reaction rate is depicted in both Figs. 2a (non-normalized) and 2b (normalized by the field area, sampled by the reactive flow). This is due to the fact that the global reaction rate (Figure 1a) exhibits a clear linear dependency on the normalized time, which is a clear indication that this rate is proportional to the field area sampled by the reactive particles. In order to emphasize that, we added a plot of the global reaction rate, normalized by the distance, passed by the reactive particles in the field. The normalized global reaction rate obtains an approximately constant value, which serves to confirm this observation.

2.2 The evolutions could be compared if (a) had its x axis in log scale, to compare with (c) and (d).

We thank the reviewer for suggesting ways to improve presentation of the manuscript. All Figure 2 subfigures have been replotted on the log-log scale to facilitate comparison (see the updated Figure 2 below). We suggest that all Figure 2 plots be plotted on a log-log scale (similar to subfigures 1c and 1d), as this would

allow to immediately discern power law relationships, that are significant for the course of the paper. Should only the X axis be put to log scale, the power law relation will not be recognizable visually.

The linearity of the global reaction rate (Figure 1a) was emphasized by specifying the slope value of 1 on the log-log scale.

Also, Figure 1d has been replaced by an updated one: for historical reasons, the heterogeneity of the hydraulic conductivity field has been examined by plotting the variance of the natural logarithm of the conductivity and not the conductivity itself. To improve clarity, we replaced Figure 1d with the plot of the variance of hyd. conductivity (NOT its natural logarithm).

[Figure]

Figure 2. influence of Peclet number on the evolution of the reactive process in the field over time: (a) Global reaction rate $\dot{R}$, (b) Normalized global reaction rate $\dot{R}_{norm}$, (c) Deviation of the mean hydraulic conductivity from initial value $\widetilde{K} - K_0$ and (d) Hydraulic conductivity variance $\sigma_{\widetilde{K}}^2$, as a function of dimensionless time $\tilde{t}$.

**2.3 Is there a prediction for the plateau of (a)?**

We thank the reviewer for this important question. The observed plateau, beginning from approximately $\tilde{t} = 1.0$, is explained by the fact that the reactive particles have sampled an

entire field at that point. As seen from the non-normalized reaction rate, it is proportional to the field area, sampled by the reactive flow. This plateau still exhibits smaller changes in conductivity due to the stochastic nature of diffusion, which manages to separate reactants and drive them out of equilibrium. Over many pore volumes, this process should amount to a significant change in the field, and, therefore, this plateau is only local, and will change at extremely long-time scales. However, this analysis is out of the scope of our study.

While this subject was referenced briefly in the manuscript (line 483 states: *We observe that the reaction increases with dimensionless time in a linear fashion, as the reactive particles sample more of the field's territory at an approximately constant mean velocity, which allows broader opportunities for reaction. Beginning from $\tilde{t} = 1.0$, the global reaction rate $\dot{R}$ becomes approximately constant, as the reactive particles have sampled an entire field at that point*), we suggest expanding this paragraph to include the reasoning that appears in our answer to the reviewer's question. The manuscript will be amended accordingly.

2.4 I find the scaling of the variance of conductivity following a reciprocal Péclet number very interesting and clean. Is there a formal reason for that? I think this important result is a bit undersold: only briefly mentioned line 533. It deserves more.

We thank the reviewer for emphasizing the importance of the Peclet scalability finding in both the parameters that characterize the evolution of the reactive process in the hydraulic conductivity field over time (Global reaction rate – see line 492 in the manuscript, Normalized global reaction rate, Mean value of the relative hydraulic conductivity – line 515 and Variance of hydraulic conductivity – line 531), as well as the thermodynamic quantities of interest (Mean normalized Shannon entropy of the transport – line 606 and Normalized Shannon entropy of the breakthrough curve (BTC) - line 640).

All parameters that characterize the evolution of the reactive process in the hydraulic conductivity field over time (Global reaction rate, Normalized global reaction rate, Mean value of the relative hydraulic conductivity and Variance of hydraulic conductivity) exhibit satisfactory scaling with the reciprocal of Peclet (see the above mentioned lines in the manuscript, notice the high values of the R square parameter that point at a good quality of the proposed fits). Here, notice that the present study employs a Lagrangian definition of the Peclet number that differs from the usual Eulerian definition (see section 2.4 in the manuscript). Another obviously necessary condition for Peclet scalability is employing normalized time $\tilde{t} = t/T_{pv}$, where $T_{pv}$ is the pore volume time.

To understand better the underlying reason for the Peclet scalability of the parameters that characterize the evolution of the reactive process in the hydraulic conductivity field over time, as well as the specific form of the Peclet scaling (such as the dependency on the reciprocal of Peclet), let us consider a simplified formulation of Advection-diffusion equation (ADE), first without and then with the reaction term.

Existence of Peclet-scalable solutions to ADE / ADRE

Consider for simplicity the 1D ADE without the reactive term:

$$\frac{\partial C}{\partial t} = -u\,\frac{\partial C}{\partial x} + D\,\frac{\partial^2 C}{\partial x^2}$$

Here, $C$ is the solute concentration, $u$ is the advective velocity field, $D$ is the diffusion coefficient and $x, t$ are the spatial and temporal coordinates. Normalize all variables as in the manuscript:

$$\tilde{C} = C/C_{ref}, \; \tilde{x} = x/L, \; \tilde{t} = t/(L/U), \; \tilde{u} = u/U$$

Here, $C_{ref}$ is the reference value for the solute concentration, $L$ is the length of the domain and $U$ is the reference advective velocity. The temporal coordinate $t$ is normalized by the pore volume time $T_{pv} = L/U$. The non-dimensionalized ADE obtained is:

$$\frac{\partial \tilde{C}}{\partial \tilde{t}} = -\tilde{u}\,\frac{\partial \tilde{C}}{\partial \tilde{x}} + \frac{1}{Pe}\frac{\partial^2 \tilde{C}}{\partial \tilde{x}^2}$$

where $Pe = L\,U/D$ is the classical definition of the Peclet number. Solutions $\tilde{C} = f(\tilde{x}, \tilde{t}, Pe)$ are possible if the normalized velocity field $\tilde{u}$ remains identical in the different Peclet number scenarios. For the reactive case, let us consider a simple scenario of adsorption / desorption reaction, for which parallels can be drawn to the dissolution / precipitation scenario in the proposed manuscript (see [Berkowitz et al], [Bear and Chang] and [Rubin et al] in the References section). ADRE with the adsorption term $G$ looks as follows:

$$\frac{\partial C}{\partial t} = -u\,\frac{\partial C}{\partial x} + D\,\frac{\partial^2 C}{\partial x^2} - G/\theta$$

where $G$ is the adsorption term and $\theta$ is the porosity of the medium. A simplest case would be to employ the linear case of the Freundlich isotherm model, where the adsorption is directly proportional to the concentration (for desorption, the sign of the reaction term should be reversed):

$$S = K_d\,C$$

where $K_d$ is the distribution coefficient. The reactive term obtains the following shape:

$$G = \rho_b\,\frac{\partial S}{\partial t} = K_d\rho_b\,\frac{\partial C}{\partial t}$$

where $\rho_b$ is the dry bulk density of the solid. After some manipulation, the ADRE for the adsorption case may be written as:

$$R\,\frac{\partial C}{\partial t} = -u\,\frac{\partial C}{\partial x} + D\,\frac{\partial^2 C}{\partial x^2}$$

where R is a non-dimensional retardation factor:

$$R = 1 + K_d\,\frac{\rho_b}{\theta}$$

The non-dimensionalized version of ADRE will be:

$$R \frac{\partial \tilde{C}}{\partial \tilde{t}} = - \tilde{u} \frac{\partial \tilde{C}}{\partial \tilde{x}} + \frac{1}{Pe} \frac{\partial^2 \tilde{C}}{\partial \tilde{x}^2}$$

In case of a minor transport-reaction interaction that leads to minor conductivity changes, as reported in the manuscript, the deviations in the velocity field can be assumed minor as well, thus $\tilde{u} = u/U \approx 1$. Therefore, we obtain:

$$R \frac{\partial \tilde{C}}{\partial \tilde{t}} = - \frac{\partial \tilde{C}}{\partial \tilde{x}} + \frac{1}{Pe} \frac{\partial^2 \tilde{C}}{\partial \tilde{x}^2}$$

and $\tilde{C} = f(\tilde{x}, \tilde{t}, Pe)$. Thus, also in the reactive case a non-dimensionalized solution $\tilde{C} = f(\tilde{x}, \tilde{t}, Pe)$ is possible. See [Berkowitz et al] for further details. The amount of reaction in this case is proportional to the concentration, thus the expression for the normalized reaction rate (equivalent to the adsorption rate in this case) $\frac{\partial \tilde{S}}{\partial \tilde{t}}$ can be obtained as follows:

$$\frac{\partial \tilde{S}}{\partial \tilde{t}} = 1/S_{ref} \frac{\partial S}{\partial \tilde{t}} = K_d C_{ref}/S_{ref} \frac{\partial \tilde{C}}{\partial \tilde{t}} = \frac{K_d C_{ref}}{R\, S_{ref}} \left( - \frac{\partial \tilde{C}}{\partial \tilde{x}} + \frac{1}{Pe} \frac{\partial^2 \tilde{C}}{\partial \tilde{x}^2} \right)$$

Here, $S_{ref}$ is the reference value for the adsorption term. To compare this parameter with the Global reaction rate in the field $\dot{R}$, we integrate $\frac{\partial \tilde{S}}{\partial \tilde{t}}$ over the 1D domain to obtain:

$$\dot{S}(\tilde{t})_{overall} = \int_0^L \frac{\partial \tilde{S}}{\partial \tilde{t}} dx = \frac{K_d C_{ref}}{R\, S_{ref}} \left( - \int_0^L \frac{\partial \tilde{C}}{\partial \tilde{x}} dx + \frac{1}{Pe} \int_0^L \frac{\partial^2 \tilde{C}}{\partial \tilde{x}^2} dx \right)$$

(recall that $\frac{\partial \tilde{C}}{\partial \tilde{x}} < 0$ for injection at x=0). For $- \frac{\partial \tilde{C}}{\partial \tilde{x}} \ll \frac{1}{Pe} \frac{\partial^2 \tilde{C}}{\partial \tilde{x}^2}$ (the advective contribution is small w.r.t the diffusive one), this result may hint at:

(a) The global reaction rate scalability with the reciprocal of Peclet.
(b) The reason why at higher Peclet numbers the Peclet dependency diminishes, and the global reaction rate approaches an approximately constant (independent of Pe) non-zero value.

Indeed, as follows from Figure 1a, the ratio between the global reaction rate for *Pe=12* and *Pe=0.38* is about 4, which may hint at the fact that the advective contribution to reaction rate is small compared to the diffusive contribution for the Peclet range in the current study, where the field alteration due to dissolution\precipitation reaction is small making it comparable to the linear Freundlich isotherm model. Since the appearance of heterogeneity, as well as deviations in the mean conductivity value, are directly connected to the amount of reaction that took place in the field, we can accept that their dependency on Peclet should be of a similar form. This summarizes the discussion on Peclet scalability findings presented in Figure 1, related to the simulations presented in the proposed manuscript.

The above results should be treated as qualitative only, since the underlying assumption of ADE/ADRE is complete mixing of reactants and Fickian transport. This is not necessarily the case in our paper, where the solute transport is modeled using Lagrangian particle tracking. To establish a formal theoretical basis for the model in the current study, tools from statistical mechanics need to be employed. Given the relative complexity of the

chemical reaction in question (dissolution/precipitation of calcite), it is well beyond the scope of this study.

**3. Figure 3**

3.1 Aspect ratio of 1 would be nice as well.

[Figure]

Figure 3. Evolution of the transport self-organization in the field for *Pe = 0.38*, as represented by the decimal logarithm of the relative non-reactive tracer concentration $\tilde{c}$, based on data obtained from the non-reactive tracer tests performed on the snapshots of the field over time: (a) $\tilde{t} = 0.25$, (b) $\tilde{t} = 0.5$, (c) $\tilde{t} = 0.75$ and (d) $\tilde{t} = 1.0$.

We thank the reviewer for suggesting ways to improve presentation of our figures. Aspect ratio has been updated. Please find attached the updated Figure 3. The snapshots of the relative non-reactive tracer concentration in the hyd. conductivity field have been taken at normalized times $\tilde{t} = 0.25, 0.5, 0.75, 1.0$ (as opposed to the original manuscript draft where snapshots at $\tilde{t} = 0.24, 0.51, 0.76, 0.96$ were used).

3.2 Does the stripe feature correspond to the conductivity K(Y) when averaged over the X direction?

We thank the reviewer for suggesting further verification of the presented results. Please find presented above the plot of X-averaged hyd. conductivity (natural logarithm of K) versus Y coordinate for normalized times $\tilde{t} = 0.25, 0.5, 0.75, 1.1$. The plot shows conductivity fluctuations, whose amplitude grows with time. Also, with time we observe that some of the inlet cells exhibit larger peaks in hyd. conductivity, this tendency growing with time (notice peaks at *7.2cm, 8.2cm* and *9cm*).

This can be explained by observing the mechanism responsible for transport self-organization in the field. The particles injected at the inlet of the field are being flux-weighted, meaning that the number of particles injected in each of the inlet cells is proportional to Darcy flux in that cell. Diffusion, treated as a stochastic agent in our model, leads to appearance of hyd. conductivity fluctuations in the initially homogeneous field. Thus, the more conductive paths receive more Darcy flux, which in turn attracts more reactive *H+* particles to them, making these paths even more conductive. This autocatalytic process leads to intensification of this phenomenon, as observed in the plot above. As shown earlier, these paths are linear in shape, with negligible tortuosity, due to the hydraulic conductivity deviations from the initial value, as well as their correlation length, being minor. This autocatalytic process leads to an increase in self-organization of the transport in the field, that is manifested by a decrease in the Shannon entropy of the transport. This further stresses the importance of entropy as a means to quantify transport self-organization due to reactive process in the field.

[Figure]

The conductivity fluctuations that correspond to $\tilde{t} = 1.0$ have been adjoined with the Non-reactive solute concentration plot for $\tilde{t} = 1.0$ in an attempt to find correlation between the X-averaged hyd. conductivity fluctuations and the stripe feature of the solute transport self-organization in the field (see plot below).

[Figure]

Non-reactive Solute Concentration (Normalized) @$T_{pv}$=1

A correlation is observed between the X-averaged hyd. conductivity fluctuations and the relative solute concentration, as higher peaks in conductivity generally correspond to higher non-reactive particle concentration (notice peaks at *7.2cm, 8.2cm* and *9cm*). It should be noted that such a correlation is possible only in the case of horizontal flowpaths in the field, as in the present study.

**4. **Figure 4: S_norm**

4.1 Caption: Snorm= (S-Smax)/Smax. Please correct parenthesis.

We sincerely thank the reviewer for noticing this error on our part. The caption for Figure 5 was corrected. Also, the Figure 4a was updated to include the Normalized Shannon entropy of the transport in the field vs. normalized X coordinate for normalized times $\tilde{t} = 0.25, 0.5, 0.75, 1.0$ (as opposed to the original manuscript draft where curves for $\tilde{t} = 0.24, 0.48, 0.72, 0.96$). Please find the updated Figure 4 attached below.

[Figure]

Figure 4. Self-organization of the transport in the field via normalized Shannon entropy $S_{norm} = (S - S_{max})/S_{max}$: (a) $S_{norm}$ vs. normalized distance from inlet $\tilde{x} = x/L$ at different dimensionless times $\tilde{t}$ for Pe = 0.38, (b) Mean value of the normalized entropy over the field $\tilde{S}_{norm}$ vs. dimensionless time $\tilde{t}$ for different values of *Pe*. Inset shows Peclet-scaled curves, obtained by assuming a power-law Peclet dependency.

4.2 The quantities presented are novel: please see further comments for figure 5.

Please see detailed comments for Figure 5 below.

**5. Figure 5: S_norm BTC**

We thank the reviewer for suggesting ways to improve clarity of the figures in the manuscript. Please find below the updated Figure 5. The breakthrough times $T_{BT}$ in the X axis of the breakthrough time histogram in Figure 5a were normalized by the pore volume time $T_{PV}$. Also, for the sake of uniformity, the histogram was presented for the normalized times $\tilde{t} = 0.25, 0.5, 0.75, 1.0$ (as opposed to the original manuscript draft where histograms for $\tilde{t} = 0.24, 0.48, 0.72, 0.96$ were used). The Normalized breakthrough curve entropy $S_{norm}^{BTC}$ in Figure 5b was recalculated based on histogram division of 1000 bins instead of 5000 bins as in the original manuscript draft. This allowed for the normalized BTC entropy to begin at -1, as expected (see detailed comments further in the document). For Figure 5c, a time direction was signified by an arrow.

[Figure]

Figure 5. Breakthrough curve self-organization: (a) Histogram of the non-reactive particle arrival times in snapshots of the reacting field at different dimensionless times $\tilde{t}$ for *Pe = 0.38*, (b) Normalized breakthrough curve entropy $S_{norm}^{BTC}$ vs. dimensionless time $\tilde{t}$ for different values of Peclet number (the inset shows Peclet-scaled curves) and (c)

Normalized field transport entropy $\tilde{S}_{norm}$ vs. Normalized breakthrough curve entropy $S_{norm}^{BTC}$ for different values of Peclet number.

**5.2** What is the tilde version of Sbtc_norm (unspecified in text)?

We thank the reviewer for capturing this error on our part. There is no tilde version of $S_{norm}^{BTC}$. The normalized Shannon entropy of the breakthrough curve is defined as $S_{norm}^{BTC} = (S_{BTC} - S_{max})/S_{max}$, and is the only BTC entropy-related parameter used in the manuscript. The manuscript will be amended accordingly.

**5.3** How come they do not start at -1, but all at -0.875? At t*=0, S = 0 and Sbtc = (0-Smax)/Smax = -1.

We sincerely thank the reviewer for capturing this error on our part. The reason that $S_{norm}^{BTC}$ in Figure 5b began from -0.875 and not from -1.0 at $\tilde{t} = 0$ was due to numerical discretization errors that occurred because the number of histogram bins *Nbins* used in BTC entropy calculation was taken to be too large (*Nbins = 5000*). Due to negligible deviations in BT times at $\tilde{t} = 0$ that appeared due to numerical round-off errors, some of the BT times fell in adjacent bins. This was fixed by using smaller *Nbins = 1000*. All subfigures in Figure 5 were replaced (notice that BTC histogram relative occurrences and BTC entropy are sensitive to *Nbins*. For example, changing *Nbins* means shifting the BTC entropy curve up/down along the Y axis).

**5.4** Hydro comparison: (also for previous figure 4 concerning Snorm). Compared to the previous plots, this quantity is very novel: could you plot it for a reference HYDRO situation for which the reader would have an intuition? For instance, a plug flow turning progressively into a Poiseuille flow: at t=0, the flow field is homogeneous, at t=0.5, the flow field is a Poiseuille flow from x=0 to x=0.5 x L and still homogeneous for x=0.5 x L to L, at t=1, the Poiseuille flow is fully developed along the X direction. (should have an analytical expression).

We sincerely thank the reviewer for suggesting ways to clarify the concepts proposed in the manuscript. The hydrological analogy suggested by the reviewer is undoubtedly interesting and worthwhile pursuing, however it is beyond the scope of our study as it is dedicated to further developing the application of thermodynamic framework concepts in the context of evolving porous medium. Moreover, to our best understanding, attempts to describe the concept of self-organization by using an analogy from the field of hydrology can be somewhat misleading, as this concept is firmly grounded in the field of thermodynamics and should be considered within that field.

To our best knowledge, to clarify the proposed thermodynamic framework it would be best to employ analogies from the world of thermodynamics, since the scenario presented in the manuscript corresponds to that of an open thermodynamic system that interacts with its surroundings due to exchange of matter and energy and

arrives at a lower-than-equilibrium entropy state (higher degree of self-organization) due to influx of power from outside (see our answer to next question for details).

In order to improve understanding of the proposed concepts by a reader, the manuscript will be amended accordingly (see our answer to the next question for details).

5.5 Thermodynamic comparison: (also for previous figure concerning Snorm).This figure (and also the former) would also be a good place to link your results for this quantity to other entropic evolution of THERMODYNAMIC transformations of reference: a Joule expansion of a perfect/real gas maybe? If not this, what thermodynamic transformation is analogous to your problem?

We sincerely thank the reviewer for suggesting ways to clarify the concepts proposed in the manuscript. The scenario presented in the manuscript corresponds to that of an open thermodynamic system that interacts with its surroundings due to exchange of matter and energy. Due to influx of power from outside, this system is kept in a non-equilibrium state that corresponds to a certain degree of internal self-organization and, thus, to a decreased entropy state. Since, according to the 2nd law of Thermodynamics, the overall entropy of the system and its surroundings cannot decrease, this decrease in entropy of a system corresponds to an increase in entropy of the surroundings by means of exporting entropy from the system to the surroundings. The export of entropy from the system to surroundings requires that work is performed on the system in order to maintain its level of self-organization. Thus, such a system exhibits a pattern characterized by the interplay between the three thermodynamic properties: entropy of the system, entropy of the surroundings and influx of power required to maintain the current level of the system's entropy. In such a system, an increase in self-organization, represented by the reduction in system's entropy, leads to an increase in the entropy of the surroundings, while requiring an increase in the influx of power into the system. In our system, these three properties are represented by the entropy of transport self-organization within the field, the entropy of the BTC and the hydraulic power, dissipated by the flow as it passes within the field.

A multitude of examples of such thermodynamic systems exists (consider for example an example of a biological life in [Schroedinger 1944] or an example of a gas laser in [Haken 1983]). *One of the simplest examples of such a system from our everyday life is a refrigerator: this system is evidently in a non-equilibrium state since the temperature within the cooling chamber is much lower than that of the surroundings. This is achieved by investing external power in the system to pump the heat outside in terms of electricity required to drive the heat pump. As a result, temperature outside the refrigerator is elevated, which corresponds to an increase in the entropy of the surroundings.* Consider an example of a refrigerator, previously unconnected to power: the temperature inside the cooling chamber equals that outside, and the refrigerator is therefore in equilibrium with its surroundings, its entropy at maximum. Now, plug the socket and the heat pump will begin working, removing heat from the cooling chamber to the surroundings. Thus, the entropy in the chamber will decrease, accompanied by an increase of

entropy of the surroundings. The decrease of temperature in the cooling chamber is made possible by investing electrical power to drive the heat pump.

To draw analogy with the dynamic development of our reactive scenario, the initially homogeneous field correspond to the cooling chamber at equilibrium with its surroundings: due to its homogeneity, the solute transport through the field is uniform, thus its entropy is at maximum (no transport self-organization). Also due the field's homogeneity, the solute injected in the field reaches outlet at the same time, therefore the BTC exhibits maximal self-organization (all solute arrives at the same time), or minimal entropy.

Following the evolution of the reactive process, heterogeneity appears in the field. This heterogeneity contributes to emergence of preferential flowpaths in the field, which can be interpreted in terms of self-organization of solute transport. The increase in heterogeneity leads to an increase in self-organization of solute transport, or a decrease in its entropy. Alongside that, a greater scatter in breakthrough times is obtained, which can be interpreted as a decrease in self-organization of the BTC, or an increase in its' entropy. Here, the entropy of the transport and the entropy of the BTC correspond to the entropy of the system and its surroundings (recall that, due to the 2nd law of Thermodynamics, a decrease in the former leads to an increase in the latter for an irreversible process). To maintain an increase in the transport self-organization in the field, made possible by the reactive process which increases the field's heterogeneity and thus creates conditions for transport self-organization, an increased hydraulic power must be supplied to the flow through the field (this increasing power is required for the flow to overcome the hydraulic resistance of the field that grows with an increase in heterogeneity). This increase in power is maintained by the applied boundary condition (b.c.) of a constant pressure drop along the field in our model, which is analogous to a perfect hydraulic pump.

Considering the classical example of Joule expansion of perfect gas, suggested by the reviewer, we note that this example considers an isolated system, so no interaction with surroundings exists in this case. Therefore, this example cannot serve as a complete analogy to our system. Since this system is isolated, its entropy can only increase, as happens when the gas expands, therefore the thermodynamic process in the system during Joule expansion follows in the opposite direction to the process described in the current study.

For further reading, please see following links [Kondepudi & Prigogine, 2014], [Haken, 1983], [Schroedinger, 1944]. For the direct application of the thermodynamic framework to flow in heterogeneous porous medium, see [Zehe et al., 2021]. See also Appendix A in the manuscript for a brief exposition of an open thermodynamic system maintained in a stationary nonequilibrium state.

5.6 Without intuition on either the X axis and the Y axis, it is hard for me to comment. Indicate the time direction on the line plots with arrows please.

We sincerely thank the reviewer for suggesting ways to clarify the concepts manifested by the figures in the manuscript. In Figure 5c the Normalized field transport entropy $\tilde{S}_{norm}$ vs. Normalized breakthrough curve entropy $S_{norm}^{BTC}$ is presented for different values of Peclet number. To build an intuition regarding this

plot, the reviewer is advised to recall that in our study the former represents the entropy of the system, while the latter represents the entropy of its surroundings. As a result of the reactive process that occurs in the field, with the passage of time the entropy of the transport decreases, while the BTC entropy increases. Thus, the direction of time corresponds to a decrease in the transport entropy and an increase in the BTC entropy.

To improve clarity, the direction of time direction in Figure 5c was signified by an arrow.

5.7 Corresponding text: Correct line 630: interpretation of values 0 and values -1 are the other way around concerning the arrival time of particles, (I think!).

We express our gratitude to the reviewer for providing valuable assistance in locating errors in the manuscript. However, in this case, to the best of our knowledge, the interpretation of the limiting values of the normalized breakthrough curve entropy is correct:

The normalized breakthrough curve entropy is defined as $S_{norm}^{BTC} = (S_{BTC} - S_{max})/S_{max}$, where $S_{BTC}$ is the breakthrough curve entropy, calculated from as explained in the manuscript and $S_{max}$ is the maximal possible entropy value, obtained in the case of a perfectly uniform distribution of arrival times (similarly to the discussion of $S_{max}$ estimation for the field transport entropy held in Section 3.2, in the case of division of the arrival times span into 5000 bins, this value will be log2 5000). Thus, $S_{norm}^{BTC}$ obtains values from 0 ($S_{BTC} = S_{max}$, no self-organization) to −1 ($S_{BTC} = 0$, maximum self-organization, corresponding to the case when all particles arrive at the outlet at the same time). Thus, the definition of the normalized entropy in the manuscript is correct.

**6. **Figure 6: Hydraulic power**

6.1 The quantity is also very novel: could you compare it to reference situation? You do it in the text, which is great: but it would really help to see it on the plots.

We thank the reviewer again for suggesting ways to clarify the concepts proposed in the manuscript. Qualitative examples have been given in the comments to the previous figure, and we shall incorporate them in the revised manuscript.

6.2 Without intuition on either the X axis or the Y axis, it is hard for me to comment. Indicate the time direction on the line plots with arrows please.

We sincerely thank the reviewer for indicating the flaws in our figure presentation which hinders the understanding of the concepts proposed in the manuscript. In Figure 6b the Normalized mean field transport entropy $S_{norm}^{BTC}$ vs. Normalized Net hydraulic power, for different values of Peclet number is presented. To build an intuition regarding this plot, the reviewer is advised to recall that in our study the former represents the entropy of the system, while the latter represents the power, invested in the system to maintain its level of self-organization, characterized by the reduced entropy.

As a result of the reactive process that occurs in the field, with the passage of time the self-organization of the transport increases, which is signified by a decrease in the entropy of the transport. This requires an increase in the hydraulic power invested in the system (recall the refrigerator example). Thus, the direction of time corresponds to a decrease in the transport entropy and an increase in the hydraulic power.

To show the magnitude of the deviations from the initially homogeneous state, the power in Figure 6a and 6b was normalized by its initial value. To improve clarity, the direction of time direction in Figure 6b was signified by an arrow.

[Figure]

Figure 6. Hydraulic power in the field and its relation to entropy: (a) Net hydraulic power vs. dimensionless time $\tilde{t}$ and (b) Normalized mean field transport entropy $S_{norm}$ vs. Net hydraulic power, for different values of Peclet number.

6.3 Also: adjust P vs Sigma2 relation employing hyd. resistances connected in parallel!

We thank the reviewer for suggesting ways to further improve validity of the concepts proposed in the manuscript. This request led to a significant revision of chapters 3.3 and 4.4, where the theoretical framework for the relation of the hydraulic power to the heterogeneity and the entropy of the system is laid, and then confirmed based on the conducted numerical simulations.

Due to the approximately linear nature of the flow paths in the field, it is reasonable to view the field as a number of hyd. conductivity elements connected in parallel. Each conductivity element represents a horizontal row of field cells (following observation from results that show the approximate linearity of the flowpaths). The constant pressure drop b.c. is applied to this system, as shown:

[Figure]

Schematics of the reactive field as a system of parallel channels

The equivalent hyd. conductivity of a single channel (elements connected in series):

$$K^{EQ}_{channel\ i} = L_x \, / \sum_j dx \cdot K_{ij}^{-1} \ (1)$$

The equivalent hyd. conductivity of the whole field (channels connected in parallel):

$$K^{EQ}_{field} = \sum_i K^{EQ}_{channel\ i} \cdot \frac{dy}{L_y} = \left(\frac{dy}{L_y}\right) \cdot \sum_i \frac{L_x}{\sum_j dx \cdot K_{ij}^{-1}} \ (2)$$

where *dx, dy* are the cell dimensions and $L_x, L_y$ are the field dimensions in the x- and y-directions, respectively. Under the assumption of parallel channels with no interaction taking place between them transversally, the total hydraulic power dissipated in the field is given by:

$$P^{TOT}_{EQ} = -\rho g \Delta h_{BC} \cdot Q_{TOT} = \rho g \left(\frac{\Delta h_{BC}}{L_x}\right)^2 L_x \, L_y \, \cdot K^{EQ}_{field} \ (3)$$

where $\Delta h_{BC}$ is the hydraulic head drop between the inlet and the outlet of the field and $Q_{TOT}$ is the total flow through the field. The validity of the parallel channel assumption was investigated by comparing the calculated equivalent total hydraulic power $P^{TOT}_{EQ}$ (obtained under this assumption) to the power calculated element-wise (without this assumption):

$$P^{TOT} = -\rho g \sum_{i,j} \Delta h_{ij} \cdot Q_{ij} \ (4)$$

where *i,j* are indices that run in x- and y-directions, respectively. Here, for both approaches, the whole field was taken into account, including the area in the immediate vicinity of the inlet where a significant dissolution event occurs. The plot below compares the Normalized hydraulic power calculated using both approaches:

[Figure]

From the comparative plot presented above, we observe that the parallel channel assumption is valid as a rule in the initial part of the reactive process, however, as the normalized time $\tilde{t}$ grows, the deviations from this assumption increase. This can be explained by the fact that $P_{EQ}^{TOT}$, calculated under the assumption of parallel channels, takes into account only the power produced by the horizontal Darcy flux component, while $P^{TOT}$, calculated in a cell-wise fashion, accounts also for the transverse (vertical) flux component. With the passage of time, due to an increase in the heterogeneity of the initially homogeneous field, flux in the transverse direction appears. With time, its contribution grows as it appears more significant.

We observe that until about $\tilde{t} = 0.4$, the parallel channel assumption appears adequate for all Peclet values. Beginning from $\tilde{t} = 0.4$, deviations from this assumption are visible. The magnitude of these deviations is in direct correlation with the Peclet number and increases with the decrease in Peclet. This can be explained again by virtue of heterogeneity, which increases with the decrease in Peclet (see Figure 1d), as well as with $\tilde{t}$.

To explain the increase in hydraulic power with $\tilde{t}$ and to relate it to the growing heterogeneity of the field, consider again the relations (2) and (3) given above. The total hydraulic power in the field, using the parallel channel assumption, is related to the hydraulic conductivity field by virtue of the equivalent hydraulic conductivity of the field (2), which can be rewritten as

$$K_{field}^{EQ} = \left(\frac{dy}{L_y}\right) \cdot \sum_i \frac{1}{R_i} \ (4)$$

where $R_i = (dx/L_x)\sum_j K_{ij}^{-1}$ is the reciprocal of the equivalent hydraulic conductivity of channel *i*, or the hydraulic resistivity of the *i-th* channel. Recall the well-known feature of the harmonic mean of some population: if the population is subjected to a mean-preserving spread (that is, its variance is increased while the

mean is kept at a constant value) – then the harmonic mean always decreases [Douglas, 2004]. Since (4) can be viewed as a reciprocal of harmonic mean of $R_i$, we expect that $K_{field}^{EQ}$ will ***increase*** with an increase in the variance of $R_i$, assuming the mean conductivity changes are minor. This is confirmed by the plot of the Variance of $R_i$ vs. dimensionless time $\tilde{t}$ given below.

[Figure]

Moreover. since $R_i$ depends directly on the hyd. conductivity field $K_{ij}$, it is reasonable to expect that $K_{field}^{EQ}$ will increase with an increase in the variance of $K_{ij}$, which signifies the emergence of heterogeneity in the initially homogeneous field. This concludes the updated model of the hydraulic conductivity field that allows to establish the relation between the hydraulic power, dissipated in the field and the degree of the field's heterogeneity.

To conclude, the increase in the hydraulic power, dissipated in the field, as well as the emergence of transport self-organization, are both the result of an increase in the heterogeneity of the field. The latter, in turn, can be viewed as a consequence of the energy invested in the field by the dissolution / precipitation reactive process.

7. At the end of the article, I am still left wondering why entropy was chosen to characterize this interesting heterogeneity of conductivity.

We thank the reviewer for pointing out topics in the manuscript that require further clarification.

The proposed manuscript investigates the dynamic coupling between the transport and the reactive process in an initially homogeneous porous medium that leads to the emergence of preferential flow paths in the medium. Following an approached suggested by [Berkowitz and Zehe, 2020], we view this phenomenon as a manifestation of transport self-organization in the field, as the spatial concentration gradients, created within the

system, distance the system from the state of perfect mixing, thus allowing for a faster and more efficient fluid transport (see Sections 1.1, 1.2 in the manuscript and references therein). As a scientific term, self-organization refers to a broad range of pattern-formation processes, occurring through interactions internal to the system, without intervention by external directing influences. This term firmly belongs to the framework of modern thermodynamics and is often given a quantitative meaning by virtue of entropy (see Section 1.2 and references therein, as well as the references at the end of this document).

Following this broadly accepted trait, the proposed manuscript employs Shannon entropy to quantify the dynamics of self-organization of the transport in the field, as well as that of the breakthrough times, resulting from the coupled transport-reactive process. For specific examples in the field of hydrology we suggest [Zehe et al. 2021], where self-organization in preferential flow paths in porous media of various degrees of spatial heterogeneity was studied and quantified in terms of Shannon entropy.

**References**

- Berkowitz, Dror, Yaron. *Contaminant Geochemistry.*
- Bear and Chang. *Modeling of Groundwater Flow and Contaminant Transport.*
- Raveh-Rubin, S., Y. Edery, I. Dror and B. Berkowitz. *Nickel migration and retention dynamics in natural soil columns,* Water Resources Research, 51, 2015.
- Kondepudi & Prigogine, *Modern Thermodynamics (2nd edition)* 2014.
- Haken, *Synergetics* 1983.
- Schrodinger, *What is Life?* 1944.
- Zehe, Loritz, Edery, Berkowitz, *Preferential Pathways for Fluid and Solutes in Heterogeneous Groundwater Systems: Self-Organization, Entropy, Work*, 2021.
- Mitchell, Douglas, *More on spreads and non-arithmetic means*, The Mathematical Gazette 88, 2004.
- Berkowitz, B. and Zehe, E.: *Surface water and groundwater: unifying conceptualization and quantification of the two "water worlds"*, Hydrology and Earth System Sciences, 24, 2020.

---

## Author Response (AR1)

**Referee Report on Shavelzon and Edery, "Shannon entropy of Transport Self-Organization Due to Dissolution/Precipitation Reaction at Varying Peclet Number in an Initially homogeneous Porous Media," submitted for publication in HESS.**

Reviewer 3: I found this manuscript very interesting. The use of Shannon entropy to measure emergent structures is nice and makes sense. The manuscript is well written and easy to follow, even for those that do not have a background in statistical physics.

We sincerely thank the author for giving valuable feedback on the manuscript draft, as well as for endorsing it for publication. Please find our reply below in red.

The way that the Shannon entropy is measured does not consider spatial correlations. It is therefore an incomplete measure of this entropy. It is equivalent to measuring the Shannon entropy of a language based solely on the frequency of its use of each letter, thus missing the information attached to larger correlated structures such as the appearance of the letter "q" almost always being followed by the letter "u." Hence, the measurement of the Shannon entropy is incomplete. It is, however, probably a good first approximation to it.

We thank the reviewer for pointing out the incompleteness of the Shannon entropy when used as a measure of transport self-organization in the field due to the fact that it does not take into account spatial correlations. However, we believe that for the specific case of an initially homogeneous field, presented in the manuscript draft, it should be accurate enough, since in this case the transport self-organization mimics the form of straight parallel channels from inlet to outlet.

Thus, estimating the spatial correlation of the solute distribution in the field will not bear any additional information in this case, as no larger-scale structures are expected to emerge in the field. However, we would like to stress an importance of taking spatial correlation into account when assessing transport self-organization in the more complex situations, such as highly heterogeneous fields where larger-scale structures can be expected to emerge.

I disagree with the arguments that internal and external entropy should add up to zero. There is viscosity in the system and total entropy, both from the emerging structures and that attached to the microscopic flow, is being produced continuously. The information entropy, which reflects macroscopic structures, is a very small part of the entire entropy budget, so the connection between the two is tenuous. The increase in conductivity as the dissolution process proceeds leads to an increase in entropy *production*. That is not the same as there is an increase in entropy as the entropy leaks away from the system in the form of heat, and this also increases.

We sincerely thank the reviewer for pointing out inaccuracies in the manuscript draft. To our best understanding, there are a few points in the manuscript draft that require further clarification:

1. The presented study does not intend to construct a complete thermodynamic formalism for the problem under investigation, such as the one that is presented in *Hansen et al 2018, 2023*. The thermodynamic framework presented in the current study is aimed at providing only the qualitative dependencies / trends between the three parameters of interest, that are the entropy of the transport self-organization, the entropy of the breakthrough curve and the hydraulic power, dissipated in the field, similar to Zehe et. al. 2021. Thus, when the relationship between the pair of entropies is considered, we only state that a decrease in the entropy of the transport self-organization leads to an increase in the breakthrough curve

entropy. Under no circumstances do we state that these two entropies should construct the entire entropy budget and, thus, give a zero sum. However, we are currently working on a complete thermodynamic formalism for the reactive flow in porous medium, where we will consider the full thermodynamic budget of all the components in our system.

Please see the following quote from the manuscript draft (l.652-657):

*…since the transport entropy within the field is decreasing for lower Peclet values and, according to the second law of Thermodynamics, the overall entropy of the system and its surroundings cannot decrease, the entropy, produced within the field needs to be exported outside (see Appendix A); this is reflected in the increased entropy of the breakthrough curve. This statement should be regarded on a qualitative level only, as, due to different spatial and temporal discretizations, changes in the Shannon entropies of the field transport and the breakthrough curves with the passage of time do not comply.*

2. To analyze the emergence of self-organization in our computational field, we consider snapshots of the field in terms of hydraulic conductivity distribution, taken at different computational times as the reactive process in the field evolves (see the description in the beginning of Section 3.1). We consider each snapshot as an open thermodynamic system and perform a non-reactive tracer test by injecting *non-reactive* solute particles at the field's inlet to quantify the transport self-organization in the field at that specific time. In such a tracer test, no changes in the hydraulic conductivity distribution occur and the non-reactive solute transport in each of these field snapshots is considered as a steady state.

To clarify these aspects, we propose the following amendments to the manuscript draft (in the beginning of Section 3.1):

*l. 336: The computational setting described in the previous section mimics the dynamics of a coupled dissolution-precipitation reactive process in a calcite porous medium, leading to an emergence of heterogeneity in an initially homogeneous field. Previous studies have shown that self-organization of the solute transport in the field is expected to emerge in such a situation in the form of preferential flow paths that lead to solute concentration gradients in the direction transverse to flow, yet the details of this self-organization emergence and evolution are critical to understanding the large-scale dynamics of the coupled reactive process in the field.*

*To analyze the emergence of self-organization in our computational field, we consider snapshots of the field in terms of hydraulic conductivity distribution, taken at different computational times as the reactive process in the field evolves. We consider each snapshot as an open thermodynamic system and perform a non-reactive tracer test by injecting non-reactive solute at the field's inlet.*

*Along the lines of Section 1.2, we argue that organized states, characterized by reduced entropy, can emerge in an open system, driven away from equilibrium due to exchange of energy or matter with surroundings. Such a system may persist in a stationary nonequilibrium state. Since, according to the second law, overall entropy cannot decrease, in such a case entropy must be exported from the system outside, leading to an increase in the entropy of its surroundings.*

*l. 459: We emphasize that the presented study does not intend to construct a complete thermodynamic formalism for the problem under investigation, such as the one that is presented in Hansen et al 2018, 2023. The thermodynamic framework presented in the current study is aimed at providing qualitative dependencies / trends between the parameters of interest, such as the entropy of the transport self-*

*organization, the hydraulic power, dissipated in the field, etc. It is our intent to arrive at a more complete thermodynamic formalism for the reactive flow in porous medium in the course of the research work.*

A recent paper considers Shannon entropy in connection with immiscible two-phase flow in porous media, defining it in a way similar to that of Equation (13) in the manuscript, see https://doi.org/10.1016/j.advwatres.2022.104336. However, rather than connecting the information entropy to the molecular entropy, these authors build a statistical mechanics around it. I believe that the same ideas would be applicable in the present manuscript.

We thank the reviewer for suggesting the work of *Hansen et al 2018, 2023* for our further consideration. The thermodynamic formalism presented therein is indeed of interest for our current research direction. We will add references to *Hansen et al 2018, 2023* to the manuscript, as both of the publications are relevant to our current research direction. We will also consider the derivation presented in these papers for our current work on the complete thermodynamic formalism for dissolution\precipitation reactive transport in porous media, as parallels can be drawn between the immiscible two-phase flow in porous medium and our setup.

Despite my disagreement with the analysis of the authors with respect to the entropy budget, I will endorse publication. However, the authors should address my concern.

We sincerely thank the reviewer for identifying the merit of our work, despite our differences in the analytical aspects of it. We aim at deriving a complete thermodynamic formalism, as suggested by the reviewer, and this work is the first building block towards this goal. We believe that we have responded faithfully and in a complete fashion to the reviewer concerns, and truly appreciate the work invested in the review process, and the enlighten comments and references.

References:

- Alex Hansen, Santanu Sinha, Dick Bedeaux, Signe Kjelstrup, Magnus Aa. Gjennestad, Morten Vassvik. *Relations Between Seepage Velocities in Immiscible, Incompressible Two-Phase Flow in Porous Media*, 2018.

- Alex Hansen, Eirik Grude Flekkøy, Santanu Sinha, Per Arne Slotte . *A statistical mechanics framework for immiscible and incompressible two-phase flow in porous media*, 2023.

- Erwin Zehe, Ralf Loritz, Yaniv Edery, and Brian Berkowitz. *Preferential pathways for fluid and solutes in heterogeneous groundwater systems: self-organization, entropy, work, 2021.*

---

## Editor Decision (ED1)

**Referee Report on Shavelzon and Ederly, "Shannon entropy of Transport Self-Organization Due to Dissolution/Precipitation Reaction at Varying Peclet Number in an Initially homogeneous Porous Media," submitted for publication in HESS.**

I found this manuscript very interesting. Th use of Shannon entropy to measure emergent structures is nice and makes sense. The manuscript is well written and easy to follow, even for those that do not have a background in statistical physics.

The way that the Shannon entropy is measured does not consider spatial correlations. It is therefore an incomplete measure of this entropy. It is equivalent to measuring the Shannon entropy of a language based solely on the frequency of its use of each letter, thus missing the information attached to larger correlated structures such as the appearance of the letter "q" almost always being followed by the letter "u." Hence, the measurement of the Shannon entropy is incomplete. It is, however, probably a good first approximation to it.

I disagree with the arguments that internal and external entropy should add up to zero. There is viscosity in the system and total entropy, both from the emerging structures and that attached to the microscopic flow, is being produced continuously. The information entropy, which reflects macroscopic structures, is a very small part of the entire entropy budget, so the connection between the two is tenuous. The increase in conductivity as the dissolution process proceeds leads to an increase in entropy *production*. That is not the same as there is an increase in entropy as the entropy leaks away from the system in the form of heat, and this also increases.

A recent paper considers Shannon entropy in connection with immiscible two-phase flow in porous media, defining it in a way similar to that of Equation (13) in the manuscript, see https://doi.org/10.1016/j.advwatres.2022.104336. However, rather than connecting the information entropy to the molecular entropy, these authors build a statistical mechanics around it. I believe that the same ideas would be applicable in the present manuscript.

Despite my disagreement with the analysis of the authors with respect to the entropy budget, I will endorse publication. However, the authors should address my concern.

---

## Author Response (AR2)

Dear Harrie-Jan Hendricks Franssen

Thank you for the time invested in our paper. We have fixed the technical issues with the COBLIS and supplement material (point 1 and 2) and replied to the second reviewer. The changes are easily tractable in the track changes file.
Best
Evgany and Yaniv

1. COBLIS - all plots were run through COBLIS and updated accordingly to allow correct interpretation by readers with various sight deficiencies.

2. Supplement was updated in accordance with:

1. Supplements will receive their own DOI (digital object identifier) and will be published online along with the article as *.zip archive or single *.pdf file.

2. Supplements will receive a title page added during the publication process including title ("Supplement of"), authors, and the correspondence email. Therefore, please avoid providing this information in the supplement.

3. Equations, figures and tables in supplements should be numbered as (S1), Fig. S5 or Table S6. Sections are numbered as S3, S3.1, and S3.1.1.

4. **The overall file size of a supplement is limited to 50 MB.** Authors of larger supplements are kindly asked to submit their files to a reliable FAIR-aligned data repository and to insert a persistent identifier, ideally a DOI, in the manuscript.

Reviewer 2:
I disagree with the internal and external entropy adding up to zero. The Shannon entropy is just a small part of the total entropy, most of it being in the molecular chaos. This molecular entropy is continuously being produced and shed to the surroundings as heat. The Shannon cannot balance it.

Reply by authors:

We accept the reviewer comment and acknowledge that there are additional processes that may affect the entropy. As such, the formulations regarding the thermodynamic context of the BTC - transport entropies relation were made less strict: instead of using the 2nd law to explain these relations, an analogy was suggested, while it was noted that it should only be taken qualitatively. Everywhere it was mentioned that these entropies are only a part of the total thermodynamic budget, so I concentrated only on the self-consistent relationship between them that we found.

---

## Author Response (AR3)

Dear Harrie-Jan Hendricks Franssen

Thank you for the time invested in our paper, and for taking care of its acceptance. We hope that we have uploaded all the needed files.

We added two insignificant,updates in the text:

1. Instead of "solid lines represent cell-wise power calculation $P_{total}$, while the dashed lines represent power, calculated based on the parallel channels model $P^{EQ}_{total}$", it is now "unmarkered lines represent cell-wise power calculation $P_{total}$, while the lines markered with circle markers represent power, calculated based on the parallel channels model $P^{EQ}_{total}$". This change appears twice in pp. 28-29. I adjusted plots to be COBLIS-compatible in the previous revision but missed the description in the text.

2. Code and data availability: Instead of "A dedicated repository, containing all codes and data, will be created on GitHub" it is now "Codes and data are available on a dedicated GitHub repository upon request from Evgeny Shavelzon (eshavelzon@campus.technion.ac.il)".

Best
Evgany and Yaniv